# LEARN2MIX: TRAINING NEURAL NETWORKS USING ADAPTIVE DATA INTEGRATION

## ABSTRACT

Accelerating model convergence in resource-constrained environments is essential for fast and efficient neural network training. This work presents *learn2mix*, a new training strategy that adaptively adjusts class proportions within batches, focusing on classes with higher error rates. Unlike classical training methods that use static class proportions, learn2mix continually adapts class proportions during training, leading to faster convergence. Empirical evaluations on benchmark datasets show that neural networks trained with learn2mix converge faster than those trained with classical approaches, achieving improved results for classification, regression, and reconstruction tasks under limited training resources and with imbalanced classes. Our empirical findings are supported by theoretical analysis.

## 1 INTRODUCTION

Deep neural networks have become essential tools across various applications of machine learning, including computer vision (Krizhevsky et al., 2012; Simonyan & Zisserman, 2014; He et al., 2016), natural language processing (Vaswani et al., 2017; Devlin et al., 2018; Radford et al., 2019; Touvron et al., 2023), and speech recognition (Hinton et al., 2012; Baevski et al., 2020). Despite their ability to learn and model complex, nonlinear relationships, deep neural networks often require substantial computational resources during training. In resource-constrained environments, this demand poses a significant challenge (Goyal et al., 2017), making the development of efficient and scalable training methodologies increasingly crucial to fully leverage the capabilities of deep neural networks.

Training deep neural networks relies on the notion of empirical risk minimization (Vapnik & Bottou, 1993), and typically involves optimizing a loss function using gradient-based algorithms (Rumelhart et al., 1986; Bottou, 2010; Kingma & Ba, 2014). Techniques such as regularization (Srivastava et al., 2014; Ioffe & Szegedy, 2015) and data augmentation (Shorten & Khoshgoftaar, 2019), learning rate scheduling, (Smith, 2017) and early stopping (Prechelt, 1998), are commonly employed to enhance generalization and prevent overfitting. However, the efficiency of the training process itself remains a critical concern, particularly in terms of convergence speed and computational resources.

Within this context, adaptive training strategies, which target enhanced generalization by modifying aspects of the training process, have emerged as promising approaches. Methods such as curriculum learning (Bengio et al., 2009; Wang et al., 2021) adjust the order and difficulty of training samples to facilitate more effective learning. These methods expand upon educational paradigms, progressively introducing more complex samples as the model proficiency increases (Graves et al., 2017). Insights from the above adaptive training strategies can also be applied to the class imbalance problem (Wang et al., 2019), where underrepresented classes are inherently harder to learn due to data scarcity (Buda et al., 2018). These methods are typically categorized into data-level methods, such as oversampling and undersampling (Chawla et al., 2002), and algorithm-level approaches, including class-balanced loss functions (Lin et al., 2017). However, developing adaptive training approaches that *accelerate* model convergence, while ensuring robustness to class imbalance, remains an open problem.

Building upon these insights, a critical aspect of training efficiency lies in the composition of batches used during stochastic gradient descent. Classical training paradigms maintain approximately fixed class proportions within each shuffled batch, mirroring the overall class distribution in the training dataset (Buda et al., 2018; Peng et al., 2019). However, this static approach fails to account for the varying levels of difficulty associated with different classes, which can hinder optimal convergence rates. For example, classes with higher error rates or those that are inherently more challenging may

Figure 1: Illustration of the learn2mix training mechanism. The class-wise composition of batches is adaptively modified during training using instantaneous class-wise error rates.

require greater emphasis during training to enhance model performance. Ignoring these nuances can lead to suboptimal learning trajectories and prolonged training periods. While existing approaches address class imbalance by adjusting sample weights or dataset resampling, they do not dynamically change the class-wise composition of batches during training via real-time performance metrics.

This observation motivates the central question of this paper: *Can we dynamically adjust the proportion of classes within batches, across training epochs, to accelerate model convergence?* Addressing this question involves developing strategies that dynamically modify the proportion of classes using real-time performance metrics, thereby directing the learning procedure towards more challenging or underperforming classes. Such adaptive batch construction has the potential to enhance convergence rates and model accuracy, providing more efficient training, especially in scenarios characterized by class imbalance or heterogeneous class difficulties (Liu et al., 2008; Ren et al., 2018).

To address these nuances, in this work, we introduce *learn2mix*, a new training strategy that dynamically modifies class proportions in batches by emphasizing classes with higher instantaneous error rates. In contrast with classical training schemes that have fixed class proportions, learn2mix continually adapts these proportions during training via real-time class-wise error metrics. This dynamic adjustment facilitates faster convergence and improved performance across various tasks, including classification, regression, and reconstruction. An illustration of the learn2mix training methodology is provided in Figure 1, demonstrating the adaptive class-wise composition of batches.

This paper is organized as follows. In Section 2, we formalize learn2mix, and prove relevant properties. In Section 3, we detail the algorithmic implementation of the learn2mix training methodology. In Section 4, we present empirical evaluations on benchmark datasets, demonstrating the efficacy of learn2mix in accelerating model convergence and enhancing performance. Finally, in Section 5, we summarize our paper. Our main contributions are outlined as follows:

1. We propose *learn2mix*, an adaptive training strategy that dynamically adjusts class proportions within batches, using class-wise error rates, to accelerate model convergence.

2. We prove that neural networks trained using *learn2mix* converge faster than those trained using classical approaches when certain properties hold, wherein the class proportions converge to a stable distribution proportional to the optimal class-wise error rates.

3. We empirically validate that neural networks trained using *learn2mix* consistently observe accelerated convergence, outperforming classical training methods in terms of convergence speed across classification, regression, and reconstruction tasks.

**Related Work.** The landscape of neural network training methods is characterized by a diverse set of approaches aiming to enhance model performance and training efficiency. Handling class imbalance has been extensively analyzed, with methods including oversampling (Chawla et al., 2002), undersampling (Tahir et al., 2012), and class-balanced loss functions (Lin et al., 2017; Ren et al., 2018) being proposed to mitigate biases towards majority classes. In parallel, curriculum learning (Bengio et al., 2009) and reinforcement learning-centric approaches (Florensa et al., 2017) have introduced ways to facilitate more effective learning trajectories. Meta-learning, or *learn2learn* methodologies (Arnold et al., 2020), including model-agnostic meta-learning (MAML) (Finn et al., 2017), focus on optimizing the learning process itself to enable rapid adaptation to new tasks, highlighting the im-

portance of adaptability in model training. Additionally, adaptive data sampling strategies (Liu et al., 2008) and boosting algorithms (Freund & Schapire, 1997) emphasize the significance of prioritizing harder or misclassified examples to improve model robustness and convergence rates. Despite these advances, most existing training methods either adjust sample weights, resample datasets, or modify the sequence of training examples without specifically altering the class proportions within batches in an adaptive manner. Our proposed *learn2mix* strategy distinguishes itself by continually adapting class proportions within these batches throughout the training process, directly targeting classes with higher error rates to accelerate convergence. This approach not only addresses class imbalance but also integrates principles from adaptive training, offering a unified framework that enhances training efficiency by accelerating model convergence across diverse tasks.

## 2 THEORETICAL RESULTS

Consider the random variables $X \in \mathbb{R}^d$ and $Y \in \mathbb{R}^k$, wherein $X$ denotes the feature vector, $Y$ are the labels, and $k$ is the number of classes. We consider the *original training dataset*, $J = \{(x_j, y_j)\}_{j=1}^N$, where $(x_j, y_j) \overset{\text{i.i.d.}}{\sim} (X, Y), \forall j \in \{1, \ldots, N\}$. The class proportions for this dataset are given by the vector of fixed-proportion mixing parameters, $\tilde{\alpha} = [\tilde{\alpha}_1, \ldots, \tilde{\alpha}_k]^T$, which reflects the distribution of classes. We define $\alpha = [\alpha_1, \ldots, \alpha_k]^T$ as a variable denoting the vector of *mixing parameters*, where $\alpha_i \in [0, 1]$ and $\sum_{i=1}^k \alpha_i = 1$. The value of $\alpha$ specifies the class proportions utilized during training, and can vary depending on the chosen training mechanism. In *classical training*, $\alpha = \alpha^t$ is constant over time and reflects the class proportions in the original training dataset, wherein $\alpha^t = \tilde{\alpha}, \forall t \in \mathbb{N}$. In *learn2mix training*, $\alpha = \alpha^t$ is time-varying, and is initialized at time $t = 0$ as $\alpha^0 = \tilde{\alpha}$.

Let $\mathcal{H} \subset \{h : \mathbb{R}^d \to \mathbb{R}^k\}$ be the class of hypothesis functions that model the relationship between $X$ and $Y$. For our empirical setting, we let $\mathcal{H}$ denote the set of neural networks that have predetermined architectures. We note $\mathcal{H}$ is fully defined by a vector of parameters, $\theta \in \mathbb{R}^m$, where $\mathcal{H} = h_\theta$ denotes a set of parameterized functions. The generalized form of the loss function for classical training and the loss function form under learn2mix training are given below.

**Definition 2.1** (Loss Function for Classical Training). *Consider $\tilde{\alpha} \in [0, 1]^k$ as the vector of fixed-proportion mixing parameters, and let $\mathcal{L}(\theta^t) \in \mathbb{R}^k$ denote the vector of class-wise losses at time $t$. The loss for classical training at time $t$ is given by:*

$$\mathcal{L}(\theta^t, \tilde{\alpha}) = \sum_{i=1}^k \tilde{\alpha}_i \mathcal{L}_i(\theta^t) = \tilde{\alpha}^T \mathcal{L}(\theta^t). \tag{1}$$

**Definition 2.2** (Loss Function for Learn2Mix Training). *Consider $\alpha^t, \alpha^{t-1} \in [0, 1]^k$ as the vector of mixing parameters at time $t$ and time $t - 1$, and let $\mathcal{L}(\theta^t), \mathcal{L}(\theta^{t-1}) \in \mathbb{R}^k$ denote the respective class-wise loss vectors at time $t$ and time $t - 1$. Consider $\gamma \in (0, 1)$ as the mixing rate. The loss for learn2mix training at time $t$ is given by:*

$$\mathcal{L}(\theta^t, \alpha^t) = \sum_{i=1}^k \alpha_i^t \mathcal{L}_i(\theta^t) = (\alpha^t)^T \mathcal{L}(\theta^t), \tag{2}$$

$$\text{Where: } \alpha^t = \alpha^{t-1} + \gamma \left( \frac{\mathcal{L}(\theta^{t-1})}{\mathbb{1}_k^T \mathcal{L}(\theta^{t-1})} - \alpha^{t-1} \right). \tag{3}$$

Let $\theta^* \in \mathbb{R}^m$ denote the parameters of the optimal hypothesis function $h_{\theta^*}$, such that $h_{\theta^*} = \mathbb{E}[Y|X]$ almost surely. In the following proposition, we demonstrate that using gradient-based optimization under learn2mix training, the parameters converge to $\theta^*$, with the mixing proportions converging to a stable distribution that reflects the relative difficulty of each class under the optimal parameters.

**Proposition 2.3.** *Let $\mathcal{L}(\theta^t), \mathcal{L}(\theta^*) \in \mathbb{R}^k$ denote the respective class-wise loss vectors for the model parameters at time $t$ and for the optimal model parameters. Suppose each class-wise loss $\mathcal{L}_i(\theta) \in \mathbb{R}$ is strongly convex in $\theta$, with strong convexity parameter $\mu_i \in \mathbb{R}_{>0}, \forall i \in \{1, \ldots, k\}$, and each class-wise loss gradient $\nabla_\theta \mathcal{L}_i(\theta) \in \mathbb{R}^m$ is Lipschitz continuous in $\theta$, having Lipschitz constant $L_i \in \mathbb{R}_{\geq 0}, \forall i \in \{1, \ldots, k\}$. Let $\mu^* = \min_{i \in \{1, \ldots, k\}} \mu_i$, $L^* = \max_{i \in \{1, \ldots, k\}} L_i$. Then, if the model parameters at time $t + 1$ are obtained via the gradient of the loss for learn2mix training, where:*

$$\theta^{t+1} = \theta^t - \eta \nabla_\theta \mathcal{L}(\theta^t, \alpha^t), \qquad \text{with:} \quad \eta \in \mathbb{R}_{>0}, \tag{4}$$

*It follows that for learning rate, $\eta \in (0, 2/L^*)$, and mixing rate, $\gamma \in (0, 1)$:*

$$\lim_{t \to \infty} \theta^t = \theta^*, \qquad and: \qquad \lim_{t \to \infty} \alpha^t = \alpha^* = \frac{\mathcal{L}(\theta^*)}{\mathbb{1}_k^T \mathcal{L}(\theta^*)}. \tag{5}$$

The complete proof of Proposition 2.3 is provided in Section A.1 of the Appendix. We now detail the convergence behavior of the learn2mix and classical training strategies, and suppose that $\alpha^{t-1} = \tilde{\alpha}$. We first present Corollary 2.4, which will be used to prove the convergence result in Proposition 2.5. This corollary leverages Lipschitz continuity and strong convexity to bound the loss gradient norm.

**Corollary 2.4.** *Let $\mathcal{L}(\theta^t) \in \mathbb{R}^k$ denote the class-wise loss vector at time $t$. Suppose each class-wise loss, $\mathcal{L}_i(\theta) \in \mathbb{R}$, is strongly convex in $\theta$, with strong convexity parameter $\mu_i \in \mathbb{R}_{>0}$, $\forall i \in \{1, \ldots, k\}$, and suppose each class-wise loss gradient $\nabla_\theta \mathcal{L}_i(\theta) \in \mathbb{R}^m$ is Lipschitz continuous in $\theta$ with Lipschitz constant $L_i \in \mathbb{R}_{\geq 0}$, $\forall i \in \{1, \ldots, k\}$. Let $\mu^* = \min_{i \in \{1, \ldots, k\}} \mu_i$, $L^* = \max_{i \in \{1, \ldots, k\}} L_i$. Then, the following condition and inequality hold, $\forall \alpha \in [0, 1]^k$ where $\sum_{i=1}^{k} \alpha_i = 1$:*

$$\frac{\mu^*}{2} \|\theta^t - \theta^*\| \leq \|\nabla_\theta \mathcal{L}(\theta^t, \alpha)\| \leq L^* \|\theta^t - \theta^*\|, \tag{6}$$

$$Wherein: \|\nabla_\theta \mathcal{L}(\theta^t, \alpha^t)\| + \|\nabla_\theta \mathcal{L}(\theta^t, \tilde{\alpha})\| \leq 2L^* \|\theta^t - \theta^*\|. \tag{7}$$

The proof of Corollary 2.4 is provided in Section A.1 of the Appendix — we note that the inequality in Eq. (7) relates the loss gradient norm under classical training with that under learn2mix training. We now present Proposition 2.5, which demonstrates that under the condition expressed in Eq. (8), updates obtained via the gradient of the loss for learn2mix training bring the model parameters closer to the optimal solution than those obtained via the gradient of the loss for classical training.

**Proposition 2.5.** *Let $\mathcal{L}(\theta^t), \mathcal{L}(\theta^*) \in \mathbb{R}^k$ denote the respective class-wise loss vectors for the model parameters at time $t$ and for the optimal model parameters. Suppose each class-wise loss, $\mathcal{L}_i(\theta) \in \mathbb{R}$ is strongly convex in $\theta$ with strong convexity parameter $\mu_i \in \mathbb{R}_{>0}$, $\forall i \in \{1, \ldots, k\}$, and each class-wise loss gradient $\nabla_\theta \mathcal{L}_i(\theta) \in \mathbb{R}^m$ is Lipschitz continuous in $\theta$, having Lipschitz constant $L_i \in \mathbb{R}_{\geq 0}$, $\forall i \in \{1, \ldots, k\}$. Moreover, suppose the loss gradient $\nabla_\theta \mathcal{L}(\theta, \alpha) \in \mathbb{R}^m$ is Lipschitz continuous in $\alpha$, having Lipschitz constant $L_\alpha \in \mathbb{R}_{\geq 0}$, and let $\mu^* = \min_{i \in \{1, \ldots, k\}} \mu_i$, $L^* = \max_{i \in \{1, \ldots, k\}} L_i$. Then, if and only if the following condition holds:*

$$\left[ \left( \frac{\mu^*}{2} - L^* \right) \|\theta^t - \theta^*\|^2 + \tilde{\alpha}^T (\mathcal{L}(\theta^t) - \mathcal{L}(\theta^*)) \right] \left[ \|\theta^t - \theta^*\| - (\mathcal{L}(\theta^t) - \mathcal{L}(\theta^*)) \right] > 0, \tag{8}$$

*It follows that for every learning rate, $\eta > 0$, there exists a mixing rate, $\gamma \in (0, \beta]$, such that:*

$$\left\| \left( \theta^t - \eta \nabla_\theta \mathcal{L}(\theta^t, \alpha^t) \right) - \theta^* \right\| \leq \left\| \left( \theta^t - \eta \nabla_\theta \mathcal{L}(\theta^t, \tilde{\alpha}) \right) - \theta^* \right\|. \tag{9}$$

*The complete formula for $\beta$ can be found in Section A.1 of the Appendix.*

The complete proof of Proposition 2.5 is provided in Section A.1 of the Appendix.

## 3 ALGORITHM

In this section, we outline our approach for training neural networks using learn2mix. The learn2mix mechanism consists of a bilevel optimization procedure, where we first update the parameters of the neural network, $\theta^t$, and then modify the mixing parameters, $\alpha^t$, using the vector of class-wise losses, $\mathcal{L}(\theta^t)$. Deriving from the original training dataset, $J$, consider $J_i = \{(x_j, y_j)\}_{j=1}^{\tilde{\alpha}_i N}$, $\forall i \in \{1, \ldots, k\}$ as each class-specific training dataset, wherein $J = \bigcup_{i=1}^{k} J_i$. These $k$ class-specific training datasets are leveraged to speed up batch construction under learn2mix, as we will later delineate. We consider the case of training a neural network using batched stochastic gradient descent, wherein for a given training epoch, $t$, the empirical loss is computed over $P = \frac{N}{M}$ total batches, where $M \in \mathbb{Z}^+$ denotes the batch size. Each batch is formed by sampling $\alpha_i^t M$ distinct examples from the $i$th class-specific training dataset, denoted as $S_i^p \subseteq J_i$, for $S^p = \biguplus_{i=1}^{k} S_i^p$. We let $\biguplus$ denote the set union operator that preserves duplicate elements. For learn2mix training, the class-wise errors, $\mathcal{L}_i(\theta^t), \forall i \in \{1, \ldots, k\}$, at training epoch $t$ are empirically computed as:

$$\mathcal{L}_i(\theta^t) = \frac{1}{P} \sum_{p=1}^{P} \left[ \frac{1}{\alpha_i^t M} \sum_{(x_j, y_j) \in S_i^p} \ell(h_{\theta^t}(x_j), y_j) \right], \tag{10}$$

---

**Algorithm 1:** Neural Network Training Under Learn2Mix

---

**Input:** $J$ (Original Training Dataset), $\theta$ (Initial NN Parameters), $\tilde{\alpha}$ (Initial Mixing Parameters),
$\quad\quad$ $\eta$ (Learning Rate), $\gamma$ (Mixing Rate), $M$ (Batch Size), $P$ (No. of Batches), $E$ (Epochs)

**Output:** $\theta$ (Trained NN Parameters)

1 **for** $i = 1, 2, \ldots k$ **do**
2 $\quad$ $J_i \leftarrow \{(x_j, y_j)\}_{j=1}^{\alpha_i N}$ $\quad$ (Initialize class-specific training datasets)
3 $\quad$ $\alpha_i \leftarrow \tilde{\alpha}_i$ $\quad$ (Initialize time-varying mixing parameters)
4 **for** $epoch = 1, 2, \ldots, E$ **do**
5 $\quad$ **for** $i = 1, 2, \ldots, k$ **do**
6 $\quad\quad$ $J_i \leftarrow \texttt{Shuffle}(J_i)$ $\quad$ (Randomly shuffle each class-specific training dataset)
7 $\quad$ **for** $p = 1, 2, \ldots, P$ **do**
8 $\quad\quad$ **for** $i = 1, 2, \ldots, k$ **do**
9 $\quad\quad\quad$ $S_i^p \leftarrow \texttt{Sample}(J_i, \alpha_i M)$ $\quad$ (Select $\alpha_i M$ distinct examples from $J_i$)
10 $\quad\quad$ $S^p \leftarrow \biguplus_{i=1}^k S_i^p$ $\quad$ (Aggregate samples to form batch $S^p$)
11 $\quad\quad$ $\mathcal{L}^p(\theta, \alpha) \leftarrow \frac{1}{M} \sum_{(x_j, y_j) \in S^p} \ell(h_\theta(x_j), y_j)$ $\quad$ (Compute loss on batch $S^p$)
12 $\quad$ $\mathcal{L}(\theta, \alpha) \leftarrow \frac{1}{P} \sum_{p=1}^P \mathcal{L}^p(\theta, \alpha)$ $\quad$ (Compute overall loss across all batches)
13 $\quad$ $\theta \leftarrow \theta - \eta \nabla_\theta \mathcal{L}(\theta, \alpha)$ $\quad$ (Update model parameters, $\theta$)
14 $\quad$ **for** $i = 1, 2, \ldots, k$ **do**
15 $\quad\quad$ $\mathcal{L}_i(\theta) \leftarrow \frac{1}{P} \sum_{p=1}^P \frac{1}{\alpha_i M} \sum_{(x_j, y_j) \in S_i^p} \ell(h_\theta(x_j), y_j)$ $\quad$ (Compute loss for class $i$)
16 $\quad$ $\alpha \leftarrow \texttt{Update\_Mixing\_Parameters}(\alpha, \mathcal{L}(\theta), \gamma)$
17 **return** $\theta$

---

Where $\ell : \mathcal{Y} \times \mathcal{Y} \to \mathbb{R}_{\geq 0}$ is a bounded per-sample loss function and computes the error between the model prediction, $h_{\theta^t}(x_j)$, and the true label, $y_j$. Accordingly, the overall empirical loss at training epoch, $t$, under the learn2mix training mechanism is given by:

$$\mathcal{L}(\theta^t, \alpha^t) = \sum_{i=1}^k \alpha_i^t \mathcal{L}_i(\theta^t) = \sum_{i=1}^k \alpha_i^t \left[ \frac{1}{P} \sum_{p=1}^P \left[ \frac{1}{\alpha_i^t M} \sum_{(x_j, y_j) \in S_i^p} \ell(h_{\theta^t}(x_j), y_j) \right] \right]. \quad (11)$$

Utilizing the empirical loss formulation from Eq. (11), we now detail the algorithmic implementation of the learn2mix training methodology on a per-sample basis, for consistency with the mathematical preliminaries in Section 2. We note that the batch processing equivalent of this procedure is a trivial extension to the domain of matrices, and was used to generate the empirical results from Section 4. Algorithm 1 outlines the primary training loop, where for each epoch, the class-specific datasets, $J_i$, are shuffled. Within each epoch, we iterate over the $P$ total batches, forming each batch by choosing $\alpha_i M$ examples from every $J_i$. The empirical loss within each batch is computed and aggregated to obtain the overall loss, $\mathcal{L}(\theta, \alpha)$, which is then used to update the neural network parameters through gradient descent. Lastly, the vector of class-wise losses, $\mathcal{L}(\theta)$, is calculated to inform the adjustment of the mixing parameters, $\alpha$, through Algorithm 2.

Algorithm 2 encapsulates the mechanism for adjusting class proportions via the mixing parameters, $\alpha$, based on the computed class-wise losses. For each class, $i \in \{1, \ldots, k\}$, the algorithm normalizes the class-wise loss, $\mathcal{L}_i(\theta)$, by the cumulative loss across classes to obtain $L_i$. The mixing parameter $\alpha_i$ is then updated by moving it towards $L_i$, with the step size controlled by the mixing rate, $\gamma$. This adaptive update ensures that classes with higher error rates receive increased attention in subsequent epochs, promoting balanced and focused learning across all classes.

Finally, we recall that during the batch construction phase, for each class, $i \in \{1, \ldots, k\}$, we select $\alpha_i M$ examples from each $J_i$ to form the subset $S_i^p \subseteq J_i$. Given the dynamic nature of the mixing parameters, $\alpha$, it is possible that this cumulative selection across batches may exhaust all the samples within a particular $J_i$ before the epoch concludes. To address this, we incorporate a cyclic selection mechanism. Formally, we define an index $\tau_i^p, \forall i \in \{1, \ldots, k\}$ and $p \in \{1, \ldots, P\}$, such that:

$$\tau_i^p = \left( \tau_i^{p-1} + \alpha_i M \right) \mod \tilde{\alpha}_i N, \quad (12)$$

---

**Algorithm 2:** Updating Mixing Parameters Using Learn2Mix

---

**Input:** $\alpha$ (Previous Mixing Parameters), $\mathcal{L}(\theta)$ (Class-wise loss vector), $\gamma$ (Mixing Rate)

**Output:** $\alpha$ (Updated Mixing Parameters)

**1 for** $i = 1, 2, \ldots, k$ **do**

   **2**    $L_i \leftarrow \frac{\mathcal{L}_i(\theta)}{\sum_{j=1}^{k} \mathcal{L}_j(\theta)}$    (Compute normalized class-wise losses)

   **3**    $\alpha_i \leftarrow \alpha_i + \gamma\left(L_i - \alpha_i\right)$    (Update mixing parameter for class $i$)

**4 return** $\alpha$

---

Where $\tau_i^0 = 0$, $\forall i \in \{1, \ldots, k\}$. Accordingly, when selecting $S_i^p$, if $\tau_i^{p-1} + \alpha_i M > \tilde{\alpha}_i N$, we wrap around to the beginning of $J_i$, effectively resetting the selection index, $\tau_i^p$ — this ensures that every example in $J_i$ is selected uniformly and repeatedly as needed throughout the training process. Thus, the selection procedure to construct $S_i^p$ can be defined as:

$$S_i^p = \biguplus_{w=0}^{\alpha_i M - 1} J_i\left[(\tau_i^{p-1} + w) \mod \tilde{\alpha}_i N\right]. \tag{13}$$

This cyclic selection procedure ensures that the required number of samples, $\alpha_i M$, for each class in every batch is maintained, even as $\alpha_i$ is dynamically updated across epochs.

## 4 Empirical Results

In this section, we present our empirical results on classification, regression, and image reconstruction tasks, across both benchmark and modified imbalanced datasets. We first present the classification results on three benchmark datasets (MNIST (Deng, 2012), Fashion-MNIST (Xiao et al., 2017), CIFAR-10 (Krizhevsky et al., 2009)), and three standard datasets with manually imbalanced classes (Imagenette (Howard, 2020), CIFAR-100 (Krizhevsky et al., 2009), and IMDB (Maas et al., 2011)). We note that for the imbalanced case, we only introduce the manual class-imbalancing to the training dataset, $J$, wherein the test dataset, $K = \{(x_j, y_j)\}_{j=1}^{N_{\text{test}}}$, is not changed. This choice ensures that the generalization performance of the network is benchmarked in a class-balanced setting. Next, for the regression task, we study two benchmark datasets with manually imbalanced classes (Wine Quality (Cortez et al., 2009), and California Housing (Géron, 2022)), and a synthetic mean estimation task, wherein the manual class-imbalancing parallels that of the classification case. Finally, we reconsider the MNIST, Fashion MNIST and CIFAR-10 datasets in the context of image reconstruction, again considering the aforementioned manual class-imbalancing procedure. A comprehensive description of these datasets and class-imbalancing strategies is provided in Section B of the Appendix.

We note that the intuition behind the application of learn2mix to regression and reconstruction tasks stems from its ability to adaptively handle different data distributions. As an example, for regression tasks involving a categorical variable taking $k$ distinct values, the samples from the original training dataset, $J$, that correspond to each of these $k$ values, can be aggregated to obtain each class-specific training dataset, $J_i$. Here, each dataset, $J_i$, represents a different underlying distribution. Paralleling the classification case, learn2mix will adaptively adjust the proportions of the class-specific datasets during training. Similarly, in the context of image reconstruction, we can treat the $k$ distinct classes being reconstructed as the values taken by a categorical variable, paralleling the regression context. This formulation supports the adaptive adjustment of class proportions under learn2mix training.

For the evaluations that follow, to ensure a fair comparison between the learn2mix training strategy and the classical training strategy, we use the same learning rate, $\eta$, and neural network architecture with initialized parameters, $\theta$, across all experiments for a given dataset. Additionally, we train each neural network through learn2mix (with mixing rate $\gamma$) and classical training for $E$ training epochs, where $E$ is dataset and task dependent[1]. In classification tasks, we also benchmark learn2mix and classical training versus 'focal training' and 'SMOTE training' (training using focal loss (Lin et al., 2017) and SMOTE oversampling (Chawla et al., 2002) — see Sections C.2 and C.3 of the Appendix for further details). The complete list of considered neural network architectures and hyperparameter choices is provided in Section C of the Appendix.

---

[1]Practically, we observe that choosing $\gamma \in [0.01, 0.5]$ yields improved performance (see empirical results).

Table 1: Test classification acc. for learn2mix (L2M), classical (CL), focal (FCL), SMOTE training.

| Epoch $t = 0.25E$ | | | | | | |
|---|---|---|---|---|---|---|
| **Dataset** | **MNIST** | **Fsh. MNIST** | **CIFAR-10** | **Imagenette** | **CIFAR-100** | **IMDB** |
| **Acc (L2M)** | $77.62_{\pm1.83}$ | $46.52_{\pm3.25}$ | $51.38_{\pm0.40}$ | $33.89_{\pm1.66}$ | $7.270_{\pm0.46}$ | $70.82_{\pm1.69}$ |
| **Acc (CL)** | $66.07_{\pm4.57}$ | $40.54_{\pm3.43}$ | $49.89_{\pm0.51}$ | $25.16_{\pm1.01}$ | $4.600_{\pm0.32}$ | $53.82_{\pm3.93}$ |
| **Acc (FCL)** | $69.92_{\pm4.71}$ | $40.59_{\pm3.42}$ | $49.59_{\pm0.70}$ | $27.63_{\pm2.15}$ | $6.836_{\pm0.27}$ | $50.89_{\pm1.10}$ |
| **Acc (SMOTE)** | $67.87_{\pm5.23}$ | $40.43_{\pm3.47}$ | $50.08_{\pm0.53}$ | $29.76_{\pm0.72}$ | $6.570_{\pm0.42}$ | $54.38_{\pm2.41}$ |
| Epoch $t = 0.5E$ | | | | | | |
| **Dataset** | **MNIST** | **Fsh. MNIST** | **CIFAR-10** | **Imagenette** | **CIFAR-100** | **IMDB** |
| **Acc (L2M)** | $85.04_{\pm1.38}$ | $60.12_{\pm1.30}$ | $56.76_{\pm0.69}$ | $43.50_{\pm0.86}$ | $12.10_{\pm0.36}$ | $76.12_{\pm2.36}$ |
| **Acc (CL)** | $82.69_{\pm1.58}$ | $54.59_{\pm3.11}$ | $55.36_{\pm0.40}$ | $33.72_{\pm1.24}$ | $8.200_{\pm0.26}$ | $72.32_{\pm3.28}$ |
| **Acc (FCL)** | $83.46_{\pm1.52}$ | $56.09_{\pm2.56}$ | $54.81_{\pm0.43}$ | $35.82_{\pm0.97}$ | $11.12_{\pm0.40}$ | $69.33_{\pm3.89}$ |
| **Acc (SMOTE)** | $82.93_{\pm1.67}$ | $54.55_{\pm3.10}$ | $54.76_{\pm0.68}$ | $38.73_{\pm0.47}$ | $10.86_{\pm0.47}$ | $66.28_{\pm1.78}$ |
| Epoch $t = E$ | | | | | | |
| **Dataset** | **MNIST** | **Fsh. MNIST** | **CIFAR-10** | **Imagenette** | **CIFAR-100** | **IMDB** |
| **Acc (L2M)** | $91.18_{\pm1.03}$ | $67.34_{\pm1.18}$ | $62.10_{\pm0.39}$ | $53.31_{\pm0.68}$ | $17.02_{\pm0.48}$ | $82.33_{\pm0.50}$ |
| **Acc (CL)** | $90.01_{\pm1.12}$ | $65.27_{\pm1.74}$ | $61.46_{\pm0.31}$ | $44.60_{\pm0.68}$ | $12.62_{\pm0.37}$ | $80.03_{\pm0.48}$ |
| **Acc (FCL)** | $90.08_{\pm1.07}$ | $66.32_{\pm1.71}$ | $61.19_{\pm0.18}$ | $45.30_{\pm0.74}$ | $14.45_{\pm0.57}$ | $79.83_{\pm0.71}$ |
| **Acc (SMOTE)** | $90.08_{\pm1.13}$ | $65.27_{\pm1.73}$ | $60.93_{\pm0.25}$ | $49.41_{\pm0.73}$ | $15.05_{\pm0.61}$ | $77.46_{\pm0.70}$ |

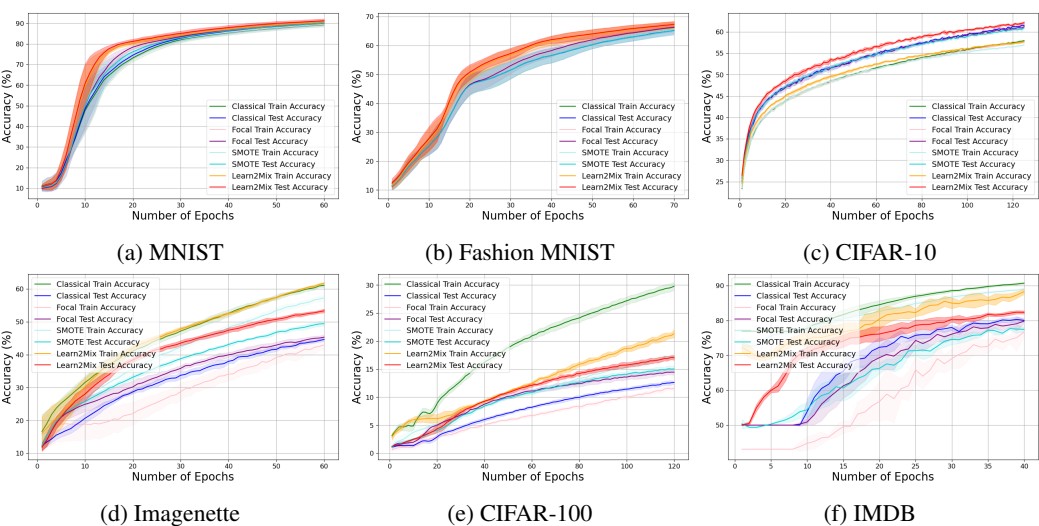

(a) MNIST  (b) Fashion MNIST  (c) CIFAR-10

(d) Imagenette  (e) CIFAR-100  (f) IMDB

Figure 2: Comparing model classification accuracies across six datasets (MNIST, Fashion MNIST, CIFAR-10, Imagenette, CIFAR-100, and IMDB Sentiment Analysis) using Cross Entropy Loss for classical training, learn2mix training, focal training, and SMOTE training. The x-axis indicates the number of elapsed training epochs, while the y-axis indicates the classification accuracy.

## 4.1 CLASSIFICATION TASKS

As illustrated in Table 1 and Figure 2, we observe a consistent trend across all tested classification benchmarks, whereby neural networks trained using learn2mix converge faster than their classically-trained, focal loss-trained, and SMOTE-trained counterparts. More concretely, we first consider the MNIST benchmark dataset. We train LeNet-5 (Lecun et al., 1998) via the Adam optimizer (Kingma & Ba, 2014) and Cross Entropy Loss for $E = 60$ epochs on MNIST, leveraging learn2mix, classical, focal, and SMOTE training. We note that the learn2mix-trained CNN achieves faster convergence, eclipsing a test accuracy of 75% after 14 epochs, whereas the respective classically-trained, focal loss-trained, and SMOTE-trained CNNs achieve this test accuracy after 20 epochs, 18 epochs, and 19 epochs. Subsequently, we consider the more challenging Fashion MNIST benchmark. We train LeNet-5 for $E = 70$ epochs with the Adam optimizer and Cross Entropy Loss on Fashion MNIST, leveraging learn2mix, classical, focal, and SMOTE training. Paralleling the MNIST case, we note that the learn2mix-trained CNN achieves faster convergence, yielding a test accuracy of 60% after

35 epochs, whereas the respective classically-trained, focal loss-trained, and SMOTE-trained CNNs achieve this test accuracy after 49 epochs, 44 epochs, and 49 epochs. The last class-balanced benchmark dataset we investigate is the CIFAR-10 dataset, which offers a greater challenge than MNIST and Fashion MNIST. We train LeNet-5 for $E = 125$ epochs using the Adam optimizer and Cross Entropy Loss on CIFAR-10, utilizing learn2mix, classical, focal, and SMOTE training. We observe that the learn2mix-trained CNN achieves faster convergence, yielding a test accuracy of $55\%$ after 50 epochs, whereas the respective classically-trained, focal loss-trained, and SMOTE-trained CNNs exceed this test accuracy after 60 epochs, 61 epochs, and 60 epochs. Cumulatively, these evaluations demonstrate the efficacy of learn2mix training even in settings with balanced classes, wherein the adaptive adjustment of class proportions accelerates convergence.

We now consider the case of benchmarking classification accuracies when the training dataset consists of imbalanced classes. We first consider the Imagenette dataset, which comprises a subset of 10 classes from ImageNet (Deng et al., 2009), and modify the training dataset such that the number of samples from each class, $i \in \{1, \ldots, k\}$, in $J$ decreases linearly. We train ResNet-18 (He et al., 2016) utilizing the Adam optimizer and Cross Entropy Loss for $E = 60$ epochs on Imagenette, via learn2mix, classical, focal, and SMOTE training. We observe that the learn2mix-trained ResNet-18 model converges faster, achieving a test accuracy of $40\%$ after 22 epochs, at which point the respective classically-trained, focal loss-trained, and SMOTE-trained ResNet-18 models have test accuracies of $30\%$, $32\%$ and $35\%$. Next, we consider the CIFAR-100 dataset, and again modify the training dataset such that the number of samples from each class, $i \in \{1, \ldots, k\}$, in $J$ decreases logarithmically. We train LeNet-5 for $E = 120$ epochs using the Adam optimizer and Cross Entropy Loss on CIFAR-100, via learn2mix, classical, focal, and SMOTE training. We see that the learn2mix-trained LeNet-5 model observes faster convergence, achieving a test accuracy of $15\%$ after 90 epochs, at which point the respective classically-trained, focal loss-trained, and SMOTE-trained CNNs have test accuracies of $11\%$ and $13.3\%$, and $13.4\%$. We further note that the $k = 100$ mixing parameters within learn2mix are a small fraction of the total model parameters, making this overhead negligible. Regarding the IMDB dataset, we modify the training dataset such that the positive class keeps $30\%$ of its original samples. We train a transformer for $E = 40$ epochs utilizing the Adam optimizer and Cross Entropy Loss on IMDB, with learn2mix, classical, focal, and SMOTE training. We find that the learn2mix-trained transformer converges faster, reaching a test accuracy of $75\%$ after 16 epochs, at which point the respective classically-trained, focal loss-trained, and SMOTE-trained transformers have test accuracies of $68\%$, $62\%$, and $61.8\%$. These experiments demonstrate the efficacy of learn2mix training over classical training and focal training in imbalanced classification settings.

We observe across the class-imbalance evaluations that learn2mix not only accelerates convergence, but also achieves a tighter alignment between training and test errors compared to classical training. This correspondence indicates reduced overfitting, as learn2mix inherently adjusts class proportions based on class-specific error rates, $L_i$. By biasing the optimization procedure away from the original class distribution and towards $L_i$, learn2mix improves the model's generalization performance. We note that this property is not unique to classification and also applies to regression and reconstruction. This behavior is empirically verified in Sections 4.2 and 4.3.

## 4.2 REGRESSION TASKS

As illustrated in Table 2 and Figure 3, we observe that learn2mix maintains accelerated convergence in the regression context, wherein all the considered datasets are class imbalanced. We first consider the synthetic Mean Estimation dataset, which comprises sets of samples gathered from $k = 4$ unique distributions and their associated means. Using the Adam optimizer and Mean Squared Error (MSE) Loss, we train a fully connected network for $E = 500$ epochs on Mean Estimation using learn2mix and classical training. We see that the learn2mix-trained neural network observes rapid convergence, achieving a test error below 2.0 after 100 epochs, at which point the classically-trained network has a test error of 13.0. For the Wine Quality dataset, we modify the training dataset such that the white wine class has $10\%$ of its original samples. Utilizing the Adam optimizer and MSE Loss, we train a fully connected network for $E = 300$ epochs on Wine Quality using learn2mix training and classical training. We observe that the learn2mix-trained neural network yields faster convergence, achieving a test error below 2.5 after 200 epochs, at which point the classically-trained network has a test error of 5.0. Finally, on the California Housing dataset, we modify the training dataset such that three of the classes have $5\%$ of their original samples. Using the Adam optimizer and MSE Loss, we train a fully connected network for $E = 1200$ epochs on California Housing using learn2mix and classical

Table 2: Test mean squared error (MSE) for learn2mix (L2M) and classical (CL) training.

| Dataset | Epoch $t = 0.25E$ | | Epoch $t = 0.5E$ | | Epoch $t = E$ | |
|---|---|---|---|---|---|---|
| | Err (L2M) | Err (CL) | Err (L2M) | Err (CL) | Err (L2M) | Err (CL) |
| **Mean Estim.** | $1.81_{\pm0.84}$ | $6.51_{\pm1.52}$ | $1.45_{\pm0.26}$ | $1.52_{\pm0.27}$ | $1.07_{\pm0.09}$ | $1.17_{\pm0.06}$ |
| **Wine Quality** | $17.7_{\pm1.64}$ | $19.8_{\pm1.51}$ | $4.26_{\pm1.55}$ | $9.72_{\pm1.94}$ | $1.75_{\pm0.21}$ | $2.03_{\pm0.18}$ |
| **Cali. Housing** | $2.52_{\pm0.68}$ | $2.95_{\pm0.67}$ | $1.33_{\pm0.32}$ | $1.82_{\pm0.39}$ | $0.77_{\pm0.08}$ | $0.99_{\pm0.10}$ |
| **MNIST** | $19.6_{\pm0.81}$ | $20.8_{\pm0.93}$ | $12.9_{\pm0.39}$ | $14.0_{\pm0.52}$ | $9.31_{\pm0.24}$ | $10.1_{\pm0.56}$ |
| **Fsh. MNIST** | $89.3_{\pm2.63}$ | $91.9_{\pm2.37}$ | $65.1_{\pm1.21}$ | $70.9_{\pm1.28}$ | $45.5_{\pm1.21}$ | $51.6_{\pm1.60}$ |
| **CIFAR-10** | $193_{\pm1.23}$ | $194_{\pm1.98}$ | $175_{\pm2.85}$ | $179_{\pm3.87}$ | $144_{\pm1.71}$ | $148_{\pm1.37}$ |

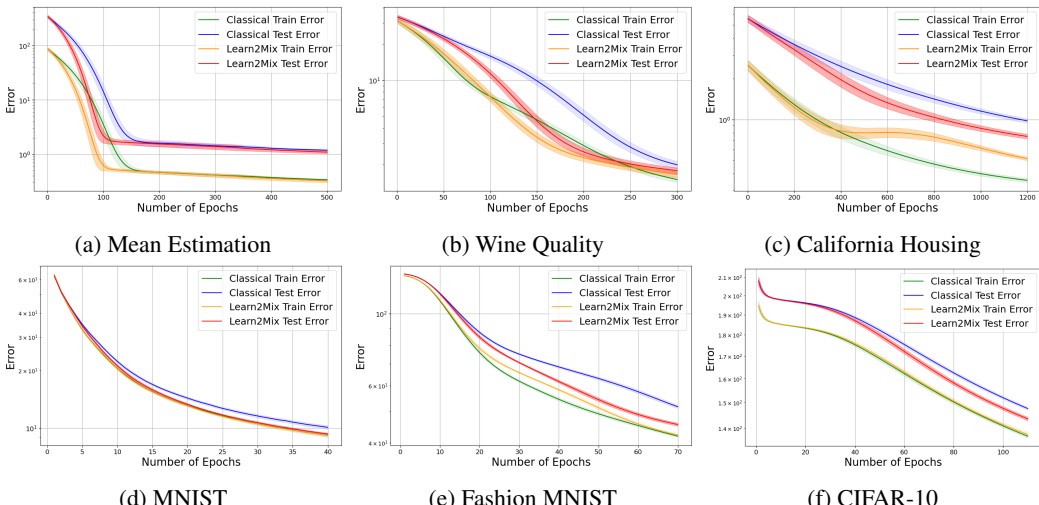

(a) Mean Estimation  (b) Wine Quality  (c) California Housing

(d) MNIST  (e) Fashion MNIST  (f) CIFAR-10

Figure 3: Comparing model performance errors across six datasets (Mean Estimation, Wine Quality, California Housing, MNIST, Fashion MNIST, and CIFAR-10) using MSE Loss for classical training and learn2mix training. The x-axis denotes the number of elapsed training epochs, while the y-axis indicates the mean squared error (MSE).

training. We again notice that the learn2mix-trained network converges faster, achieving a test error below $0.8$ after 1200 epochs, whereas the classically-trained network has a test error of $0.99$. These empirical evaluations support our previous intuition pertaining to the extension of learn2mix to class-imbalanced regression settings, wherein we observe faster convergence and reduced overfitting.

## 4.3 IMAGE RECONSTRUCTION TASKS

Per Table 2 and Figure 3, we note that the class-imbalanced image reconstruction tasks also observe faster convergence using learn2mix. For the MNIST case, we modify the training dataset such that half of the classes retain $20\%$ of their original samples. Leveraging the Adam optimizer and MSE Loss, we train an autoencoder for $E = 40$ epochs on MNIST using learn2mix and classical training. We observe that the learn2mix-trained autoencoder exhibits improved convergence, achieving a test error below $1.0$ after 35 epochs, which the classically-trained autoencoder achieves after 40 epochs. Correspondingly, for Fashion MNIST, we modify the training dataset such that half of the classes retain $20\%$ of their original samples (paralleling MNIST). Using the Adam optimizer and MSE Loss, we train an autoencoder for $E = 70$ epochs on Fashion MNIST, leveraging learn2mix and classical training. We observe that the learn2mix-trained autoencoder converges faster, achieving a test error below $54.0$ after 50 epochs, which the classically-trained autoencoder achieves after 65 epochs. We also consider CIFAR-10, wherein we modify the training dataset such that all but two classes retain $20\%$ of their original samples. Utilizing the Adam optimizer and MSE Loss, we train an autoencoder for $E = 110$ epochs on CIFAR-10, leveraging learn2mix and classical training. We observe that the learn2mix-trained autoencoder also converges faster and achieves a test error below $148.0$ after 100

epochs, which the classically-trained autoencoder achieves after 110 epochs. Cumulatively, these empirical evaluations demonstrate the improved performance yielded by learn2mix trained models over classically trained models in limited and constrained training regimes.

## 5 CONCLUSION

In this work, we introduced *learn2mix*, an adaptive training strategy that dynamically modifies class proportions in batches via real-time class-wise error rates to accelerate neural network convergence. We formalized the learn2mix mechanism through a bilevel optimization framework, and outlined its theoretical advantages in aligning class proportions with optimal error rates. Empirical evaluations across classification, regression, and reconstruction tasks on both balanced and imbalanced datasets confirmed that learn2mix not only accelerates convergence compared to classical training methods, but also reduces overfitting in the presence of class-imbalances. As a consequence, models trained with learn2mix achieved improved performance in constrained training regimes and also maintained closer alignment between training and test errors. Our findings underscore the potential of dynamic batch composition strategies in optimizing neural network training, paving the way for more efficient and robust machine learning models in resource-constrained environments.

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

# A APPENDIX

## A.1 PROOFS OF THE THEORETICAL RESULTS

In this section, we present the proofs of the theoretical results outlined in the main text.

**Proposition 2.3.** *Let $\mathcal{L}(\theta^t), \mathcal{L}(\theta^*) \in \mathbb{R}^k$ denote the respective class-wise loss vectors for the model parameters at time $t$ and for the optimal model parameters. Suppose each class-wise loss $\mathcal{L}_i(\theta) \in \mathbb{R}$ is strongly convex in $\theta$, with strong convexity parameter $\mu_i \in \mathbb{R}_{>0}$, $\forall i \in \{1, \ldots, k\}$, and each class-wise loss gradient $\nabla_\theta \mathcal{L}_i(\theta) \in \mathbb{R}^m$ is Lipschitz continuous in $\theta$, having Lipschitz constant $L_i \in \mathbb{R}_{\geq 0}$, $\forall i \in \{1, \ldots, k\}$. Let $\mu^* = \min_{i \in \{1,\ldots,k\}} \mu_i$, $L^* = \max_{i \in \{1,\ldots,k\}} L_i$. Then, if the model parameters at time $t+1$ are obtained via the gradient of the loss for learn2mix training, where:*

$$\theta^{t+1} = \theta^t - \eta \nabla_\theta \mathcal{L}(\theta^t, \alpha^t), \qquad \text{with:} \quad \eta \in \mathbb{R}_{>0}, \tag{14}$$

*It follows that for learning rate, $\eta \in (0, 2/L^*)$, and mixing rate, $\gamma \in (0, 1)$:*

$$\lim_{t \to \infty} \theta^t = \theta^*, \qquad \text{and:} \quad \lim_{t \to \infty} \alpha^t = \alpha^* = \frac{\mathcal{L}(\theta^*)}{\mathbb{1}_k^T \mathcal{L}(\theta^*)}. \tag{15}$$

*Proof.* We begin by recalling that $\mathcal{L}_i(\theta)$ is strongly convex in $\theta$ with strong convexity parameter $\mu_i$, $\forall i \in \{1, \ldots, k\}$. Accordingly, $\forall \alpha \in [0,1]^k$, with $\sum_{i=1}^k \alpha_i = 1$, the loss function $\mathcal{L}(\theta, \alpha)$ is strongly convex in $\theta$ with parameter, $\mu' \in \mathbb{R}_{>0}$, which is lower bounded by $\mu^* \in \mathbb{R}_{>0}$, as per Eq. (16).

$$\mu' \geq \mu^* > 0, \qquad \text{where:} \quad \mu^* = \min_{i \in \{1,\ldots,k\}} \mu_i, \qquad \text{and:} \quad \mu' = \sum_{i=1}^k \alpha_i \mu_i. \tag{16}$$

We note that this lower bound on the strong convexity parameter, $\mu' \geq \mu^*$, holds independently of $\alpha$. Now, recall that $\nabla_\theta \mathcal{L}_i(\theta)$, is Lipschitz continuous in $\theta$ with Lipschitz constant $L_i$, $\forall i \in \{1, \ldots, k\}$. Accordingly, $\forall \alpha \in [0,1]^k$, where $\sum_{i=1}^k \alpha_i = 1$, the loss gradient $\nabla_\theta \mathcal{L}(\theta, \alpha)$ is Lipschitz continuous in $\theta$ with Lipschitz constant, $L' \in \mathbb{R}_{\geq 0}$, which is upper bounded by $L^* \in \mathbb{R}_{\geq 0}$, as per Eq. (17).

$$L^* \geq L' \geq 0, \qquad \text{where:} \quad L^* = \max_{i \in \{1,\ldots,k\}} L_i, \qquad \text{and:} \quad L' = \sum_{i=1}^k \alpha_i L_i. \tag{17}$$

We affirm that this upper bound on the Lipschitz constant, $L' \leq L^*$, holds independently of $\alpha$. Now, suppose that $\alpha = \alpha^t$, where $\mathcal{L}(\theta, \alpha^t)$ is strongly convex in $\theta$ with parameter $\mu' \geq \mu^*$ and $\nabla_\theta \mathcal{L}(\theta, \alpha^t)$ is Lipschitz continuous in $\theta$ with constant $L' \leq L^*$. Let $\rho = \max\{|1 - \eta\mu^*|, |1 - \eta L^*|\}$. By the gradient descent convergence theorem, for learning rate, $\eta \in (0, 2/L^*)$, it follows that:

$$\lim_{t \to \infty} \|\theta^t - \theta^*\| \leq \lim_{t \to \infty} \rho^t \|\theta^0 - \theta^*\| = \|\theta^0 - \theta^*\| \lim_{t \to \infty} \rho^t = 0. \tag{18}$$

Therefore, $\lim_{t \to \infty} \theta^t = \theta^*$. Let $\beta^{t-1} = \mathcal{L}(\theta^{t-1})/\left[\mathbb{1}_k^T \mathcal{L}(\theta^{t-1})\right]$, wherein $\beta^{t-1} \in [0,1]^k$. Unrolling the recurrence relation from Eq. (5) and expressing it in terms of $\beta^{t-1}$, we obtain:

$$\alpha^t = (1-\gamma)^t \alpha^0 + \gamma \sum_{l=0}^{t-1} (1-\gamma)^{t-1-l} \beta^l. \tag{19}$$

Taking the limit and re-indexing the summation using $n = t - 1 - l$ and $l = t - 1 - n$, we obtain:

$$\lim_{t \to \infty} \alpha^t = \lim_{t \to \infty} \left[(1-\gamma)^t \alpha^0\right] + \lim_{t \to \infty} \left[\gamma \sum_{n=0}^{t-1} (1-\gamma)^n \beta^{t-1-n}\right] \tag{20}$$

$$= \mathbf{0}_k + \gamma \lim_{t \to \infty} \left[\sum_{n=0}^{t-1} (1-\gamma)^n \beta^{t-1-n}\right]. \tag{21}$$

We proceed with the steps to invoke the dominated convergence theorem. We note that for fixed $n$:

$$\lim_{t \to \infty} \left[(1-\gamma)^n \beta^{t-1-n}\right] = (1-\gamma)^n \lim_{t \to \infty} \left[\frac{\mathcal{L}(\theta^{t-1})}{\mathbb{1}_k^T \mathcal{L}(\theta^{t-1})}\right] = (1-\gamma)^n \frac{\mathcal{L}(\theta^*)}{\mathbb{1}_k^T \mathcal{L}(\theta^*)}. \tag{22}$$

Now, consider $g(n) = (1 - \gamma)^n$. For this choice of $g(n)$, we have that:

$$\|(1 - \gamma)^n \beta^{t-1-n}\| \leq (1 - \gamma)^n \|\beta^{t-1-n}\| \leq g(n), \ \forall t, n \in \mathbb{N} \tag{23}$$

$$\sum_{n=0}^{\infty} g(n) = \sum_{n=0}^{\infty} (1 - \gamma)^n = \frac{1}{1 - (1 - \gamma)} = \frac{1}{\gamma} < \infty. \tag{24}$$

We now invoke the dominated convergence theorem. Recalling Eq. (21), we observe that:

$$\lim_{t \to \infty} \alpha^t = \gamma \lim_{t \to \infty} \left[ \sum_{n=0}^{t-1} (1 - \gamma)^n \beta^{t-1-n} \right] \tag{25}$$

$$= \gamma \sum_{n=0}^{\infty} (1 - \gamma)^n \lim_{t \to \infty} \beta^{t-1-n} = \gamma \sum_{n=0}^{\infty} (1 - \gamma)^n \frac{\mathcal{L}(\theta^*)}{\mathbb{1}_k^T \mathcal{L}(\theta^*)} \tag{26}$$

$$= (\gamma) \left( \frac{1}{\gamma} \right) \frac{\mathcal{L}(\theta^*)}{\mathbb{1}_k^T \mathcal{L}(\theta^*)} = \frac{\mathcal{L}(\theta^*)}{\mathbb{1}_k^T \mathcal{L}(\theta^*)} = \alpha^*. \tag{27}$$

Therefore, $\lim_{t \to \infty} \alpha^t = \alpha^* = \mathcal{L}(\theta^*) / \left[ \mathbb{1}_k^T \mathcal{L}(\theta^*) \right]$. Cumulatively, for $\eta \in (0, 2/L^*)$ and $\gamma \in (0, 1)$, under learn2mix training, $\lim_{t \to \infty} \theta^t = \theta^*$, and $\lim_{t \to \infty} \alpha^t = \alpha^* = \mathcal{L}(\theta^*) / \left[ \mathbb{1}_k^T \mathcal{L}(\theta^*) \right]$. □

**Corollary 2.4.** *Let $\mathcal{L}(\theta^t) \in \mathbb{R}^k$ denote the class-wise loss vector at time $t$. Suppose each class-wise loss, $\mathcal{L}_i(\theta) \in \mathbb{R}$, is strongly convex in $\theta$, with strong convexity parameter $\mu_i \in \mathbb{R}_{>0}$, $\forall i \in \{1, \ldots, k\}$, and suppose each class-wise loss gradient $\nabla_\theta \mathcal{L}_i(\theta) \in \mathbb{R}^m$ is Lipschitz continuous in $\theta$ with Lipschitz constant $L_i \in \mathbb{R}_{\geq 0}$, $\forall i \in \{1, \ldots, k\}$. Let $\mu^* = \min_{i \in \{1, \ldots, k\}} \mu_i$, $L^* = \max_{i \in \{1, \ldots, k\}} L_i$. Then, the following condition and inequality hold, $\forall \alpha \in [0, 1]^k$ where $\sum_{i=1}^k \alpha_i = 1$:*

$$\frac{\mu^*}{2} \|\theta^t - \theta^*\| \leq \|\nabla_\theta \mathcal{L}(\theta^t, \alpha)\| \leq L^* \|\theta^t - \theta^*\|, \tag{28}$$

$$\textit{Wherein: } \|\nabla_\theta \mathcal{L}(\theta^t, \alpha^t)\| + \|\nabla_\theta \mathcal{L}(\theta^t, \tilde{\alpha})\| \leq 2L^* \|\theta^t - \theta^*\|. \tag{29}$$

*Proof.* We begin by recalling that $\mathcal{L}_i(\theta)$ is strongly convex in $\theta$ with strong convexity parameter $\mu_i$, $\forall i \in \{1, \ldots, k\}$. Accordingly, $\forall \alpha \in [0, 1]^k$, with $\sum_{i=1}^k \alpha_i = 1$, the loss function $\mathcal{L}(\theta, \alpha)$ is strongly convex in $\theta$ with parameter, $\mu' \in \mathbb{R}_{>0}$, which is lower bounded by $\mu^* \in \mathbb{R}_{>0}$, as per Eq. (30).

$$\mu' \geq \mu^* > 0, \quad \text{where:} \quad \mu^* = \min_{i \in \{1, \ldots, k\}} \mu_i, \quad \text{and:} \quad \mu' = \sum_{i=1}^k \alpha_i \mu_i. \tag{30}$$

Now, recall that $\nabla_\theta \mathcal{L}_i(\theta)$, is Lipschitz continuous in $\theta$ with Lipschitz constant $L_i$, $\forall i \in \{1, \ldots, k\}$. Accordingly, $\forall \alpha \in [0, 1]^k$, where $\sum_{i=1}^k \alpha_i = 1$, the loss gradient $\nabla_\theta \mathcal{L}(\theta, \alpha)$ is Lipschitz continuous in $\theta$ with Lipschitz constant, $L' \in \mathbb{R}_{\geq 0}$, which is upper bounded by $L^* \in \mathbb{R}_{\geq 0}$, as per Eq. (31).

$$L^* \geq L' \geq 0, \quad \text{where:} \quad L^* = \max_{i \in \{1, \ldots, k\}} L_i, \quad \text{and:} \quad L' = \sum_{i=1}^k \alpha_i L_i. \tag{31}$$

Note that $\nabla_\theta \mathcal{L}(\theta^*, \alpha) = \mathbf{0}_m$. Since $\mathcal{L}(\theta, \alpha)$ is strongly convex in $\theta$, the following inequalities hold:

$$\mathcal{L}(\theta^t, \alpha) - \mathcal{L}(\theta^*, \alpha) \geq \nabla_\theta \mathcal{L}(\theta^*, \alpha)^T (\theta^t - \theta^*) + \frac{\mu'}{2} \|\theta^t - \theta^*\|^2 = \frac{\mu'}{2} \|\theta^t - \theta^*\|^2, \tag{32}$$

$$\mathcal{L}(\theta^t, \alpha) - \mathcal{L}(\theta^*, \alpha) \leq \nabla_\theta \mathcal{L}(\theta^t, \alpha)^T (\theta^t - \theta^*) \leq \|\nabla_\theta \mathcal{L}(\theta^t, \alpha)\| \|\theta^t - \theta^*\|. \tag{33}$$

Combining Eq. (32) and Eq. (33), and recalling Eq. (30), we obtain the following inequality:

$$\|\nabla_\theta \mathcal{L}(\theta^t, \alpha)\| \geq \frac{\mathcal{L}(\theta^t, \alpha) - \mathcal{L}(\theta^*, \alpha)}{\|\theta^t - \theta^*\|} \geq \frac{\mu^*}{2} \|\theta^t - \theta^*\|. \tag{34}$$

Furthermore, since $\nabla_\theta \mathcal{L}(\theta, \alpha)$ is Lipschitz continuous in $\theta$ and recalling Eq. (31), it follows that:

$$\|\nabla_\theta \mathcal{L}(\theta^t, \alpha) - \nabla_\theta \mathcal{L}(\theta^*, \alpha)\| \leq L' \|\theta^t - \theta^*\| \implies \|\nabla_\theta \mathcal{L}(\theta^t, \alpha)\| \leq L^* \|\theta^t - \theta^*\|. \tag{35}$$

Altogether, combining Eq. (34) and Eq. (35), we arrive at the final inequality:

$$\frac{\mu^*}{2} \|\theta^t - \theta^*\| \leq \|\nabla_\theta \mathcal{L}(\theta^t, \alpha)\| \leq L^* \|\theta^t - \theta^*\|. \tag{36}$$

Furthermore, since Eq. (35) holds $\forall \alpha \in [0, 1]^k$ where $\sum_{i=1}^k \alpha_i = 1$, it follows that:

$$\|\nabla_\theta \mathcal{L}(\theta^t, \alpha^t)\| + \|\nabla_\theta \mathcal{L}(\theta^t, \tilde{\alpha})\| \leq 2L^* \|\theta^t - \theta^*\|. \tag{37}$$

□

**Proposition 2.5.** *Let $\mathcal{L}(\theta^t), \mathcal{L}(\theta^*) \in \mathbb{R}^k$ denote the respective class-wise loss vectors for the model parameters at time $t$ and for the optimal model parameters. Suppose each class-wise loss, $\mathcal{L}_i(\theta) \in \mathbb{R}$ is strongly convex in $\theta$ with strong convexity parameter $\mu_i \in \mathbb{R}_{>0}$, $\forall i \in \{1, \ldots, k\}$, and each class-wise loss gradient $\nabla_\theta \mathcal{L}_i(\theta) \in \mathbb{R}^m$ is Lipschitz continuous in $\theta$, having Lipschitz constant $L_i \in \mathbb{R}_{\geq 0}$, $\forall i \in \{1, \ldots, k\}$. Moreover, suppose the loss gradient $\nabla_\theta \mathcal{L}(\theta, \alpha) \in \mathbb{R}^m$ is Lipschitz continuous in $\alpha$, having Lipschitz constant $L_\alpha \in \mathbb{R}_{\geq 0}$, and let $\mu^* = \min_{i \in \{1, \ldots, k\}} \mu_i$, $L^* = \max_{i \in \{1, \ldots, k\}} L_i$. Then, if and only if the following condition holds:*

$$\left[ \left( \frac{\mu^*}{2} - L^* \right) \|\theta^t - \theta^*\|^2 + \tilde{\alpha}^T (\mathcal{L}(\theta^t) - \mathcal{L}(\theta^*)) \right] \left[ \|\theta^t - \theta^*\| - (\mathcal{L}(\theta^t) - \mathcal{L}(\theta^*)) \right] > 0, \quad (38)$$

*It follows that for every learning rate, $\eta > 0$, there exists a mixing rate, $\gamma \in (0, \beta]$, such that:*

$$\left\| \left( \theta^t - \eta \nabla_\theta \mathcal{L}(\theta^t, \alpha^t) \right) - \theta^* \right\| \leq \left\| \left( \theta^t - \eta \nabla_\theta \mathcal{L}(\theta^t, \tilde{\alpha}) \right) - \theta^* \right\|, \quad (39)$$

$$\text{Where: } \beta = \frac{\left( \frac{\mu^*}{2} - L^* \right) \|\theta^t - \theta^*\|^2 + \tilde{\alpha}^T (\mathcal{L}(\theta^t) - \mathcal{L}(\theta^*))}{\eta L_\alpha L^* \left\| \frac{\mathcal{L}(\theta^{t-1})}{\mathbb{1}_k^T \mathcal{L}(\theta^{t-1})} - \tilde{\alpha} \right\| \left[ \|\theta^t - \theta^*\| - (\mathcal{L}(\theta^t) - \mathcal{L}(\theta^*)) \right]} \quad (40)$$

*Proof.* We note that for all subsequent derivations, $\mathcal{F}(\theta^t, \theta^*, \eta, \alpha^t) = \|(\theta^t - \eta \nabla_\theta \mathcal{L}(\theta^t, \alpha^t)) - \theta^*\|$, and $\mathcal{G}(\theta^t, \theta^*, \eta, \tilde{\alpha}) = \|(\theta^t - \eta \nabla_\theta \mathcal{L}(\theta^t, \tilde{\alpha})) - \theta^*\|$, where $\alpha^{t-1} = \tilde{\alpha}$. We begin by observing that:

$$\left[ \mathcal{F}(\theta^t, \theta^*, \eta, \alpha^t) \right]^2 = \|\theta^t - \theta^*\|^2 - 2\eta(\theta^t - \theta^*)^T \nabla_\theta \mathcal{L}(\theta^t, \alpha^t) + \eta^2 \|\nabla_\theta \mathcal{L}(\theta^t, \alpha^t)\|^2, \quad (41)$$

$$\left[ \mathcal{F}(\theta^t, \theta^*, \eta, \tilde{\alpha}) \right]^2 = \|\theta^t - \theta^*\|^2 - 2\eta(\theta^t - \theta^*)^T \nabla_\theta \mathcal{L}(\theta^t, \tilde{\alpha}) + \eta^2 \|\nabla_\theta \mathcal{L}(\theta^t, \tilde{\alpha})\|^2. \quad (42)$$

Accordingly, the difference between $\left[ \mathcal{F}(\theta^t, \theta^*, \eta, \alpha^t) \right]^2$ and $\left[ \mathcal{G}(\theta^t, \theta^*, \eta, \tilde{\alpha}) \right]^2$ is given by:

$$\left[ \mathcal{F}(\theta^t, \theta^*, \eta, \alpha^t) \right]^2 - \left[ \mathcal{G}(\theta^t, \theta^*, \eta, \tilde{\alpha}) \right]^2 = -2\eta \left[ (\theta^t - \theta^*)^T (\nabla_\theta \mathcal{L}(\theta^t, \alpha^t) - \nabla_\theta \mathcal{L}(\theta^t, \tilde{\alpha})) \right] \quad (43)$$
$$+ \eta^2 \left[ \|\nabla_\theta \mathcal{L}(\theta^t, \alpha^t)\|^2 - \|\nabla_\theta \mathcal{L}(\theta^t, \tilde{\alpha})\|^2 \right].$$

Consequently, suppose that $\mathcal{H}(\theta^t, \theta^*, \eta, \tilde{\alpha}, \alpha^t) = 2\eta \left[ (\theta^t - \theta^*)^T (\nabla_\theta \mathcal{L}(\theta^t, \alpha^t) - \nabla_\theta \mathcal{L}(\theta^t, \tilde{\alpha})) \right]$, and let $\mathcal{J}(\theta^t, \eta, \tilde{\alpha}, \alpha^t) = \eta^2 \left[ \|\nabla_\theta \mathcal{L}(\theta^t, \alpha^t)\|^2 - \|\nabla_\theta \mathcal{L}(\theta^t, \tilde{\alpha})\|^2 \right]$. Suppose the loss gradient, $\nabla_\theta \mathcal{L}(\theta, \alpha)$, is Lipschitz continuous in $\alpha$ with Lipschitz constant, $L_\alpha$. We now upper bound $\mathcal{J}(\theta^t, \eta, \tilde{\alpha}, \alpha^t)$:

$$\mathcal{J}(\theta^t, \eta, \alpha, \alpha^t) = \eta^2 \left[ \nabla_\theta \mathcal{L}(\theta^t, \alpha^t) - \nabla_\theta \mathcal{L}(\theta^t, \tilde{\alpha}) \right]^T \left[ \nabla_\theta \mathcal{L}(\theta^t, \alpha^t) + \nabla_\theta \mathcal{L}(\theta^t, \tilde{\alpha}) \right]$$

$$\leq \|\nabla_\theta \mathcal{L}(\theta^t, \alpha^t) - \nabla_\theta \mathcal{L}(\theta^t, \tilde{\alpha})\| \|\nabla_\theta \mathcal{L}(\theta^t, \alpha^t) + \nabla_\theta \mathcal{L}(\theta^t, \tilde{\alpha})\| \quad (44)$$

$$\leq 2\eta^2 L_\alpha \|\alpha^t - \tilde{\alpha}\| \left[ \|\nabla_\theta \mathcal{L}(\theta^t, \alpha^t)\| + \|\nabla_\theta \mathcal{L}(\theta^t, \tilde{\alpha})\| \right] \quad (45)$$

$$\leq 2\eta^2 L_\alpha L^* \|\alpha^t - \tilde{\alpha}\| \|\theta^t - \theta^*\| \quad (46)$$

$$= 2\eta^2 L_\alpha L^* \left\| \tilde{\alpha} + \gamma \left( \frac{\mathcal{L}(\theta^{t-1})}{\mathbb{1}_k^T \mathcal{L}(\theta^{t-1})} - \tilde{\alpha} \right) - \tilde{\alpha} \right\| \|\theta^t - \theta^*\| \quad (47)$$

$$= 2\eta^2 L_\alpha L^* \gamma \left\| \frac{\mathcal{L}(\theta^{t-1})}{\mathbb{1}_k^T \mathcal{L}(\theta^{t-1})} - \tilde{\alpha} \right\| \|\theta^t - \theta^*\|. \quad (48)$$

We note that this upper bound follows from the Cauchy-Schwarz inequality and Corollary 2.4. We proceed by lower bounding $\mathcal{H}(\theta^t, \theta^*, \eta, \tilde{\alpha}, \alpha^t)$:

$$\mathcal{H}(\theta^t, \theta^*, \eta, \tilde{\alpha}, \alpha^t) = 2\eta \left[ (\theta^t - \theta^*)^T \nabla_\theta \mathcal{L}(\theta^t, \alpha^t) - (\theta^t - \theta^*)^T \nabla_\theta \mathcal{L}(\theta^t, \tilde{\alpha}) \right] \quad (49)$$

$$\geq 2\eta \left[ (\theta^t - \theta^*)^T \nabla_\theta \mathcal{L}(\theta^t, \alpha^t) - \|\theta^t - \theta^*\| \|\nabla_\theta \mathcal{L}(\theta^t, \tilde{\alpha})\| \right] \quad (50)$$

$$\geq 2\eta \left[ (\theta^t - \theta^*)^T \nabla_\theta \mathcal{L}(\theta^t, \alpha^t) - L^* \|\theta^t - \theta^*\|^2 \right] \quad (51)$$

$$= 2\eta \left[ \frac{\mu^*}{2} \|\theta^t - \theta^*\|^2 + \mathcal{L}(\theta^t, \alpha^t) - \mathcal{L}(\theta^*, \alpha^t) - L^* \|\theta^t - \theta^*\|^2 \right] \quad (52)$$

$$= 2\eta \left[ \left( \frac{\mu^*}{2} - L^* \right) \|\theta^t - \theta^*\|^2 + \tilde{\alpha}^T (\mathcal{L}(\theta^t) - \mathcal{L}(\theta^*)) \right.$$

$$\left. + \gamma \left( \frac{\mathcal{L}(\theta^{t-1})}{\mathbb{1}^T \mathcal{L}(\theta^{t-1})} - \tilde{\alpha} \right)^T (\mathcal{L}(\theta^t) - \mathcal{L}(\theta^*)) \right]. \quad (53)$$

We note that this lower bound also follows from the Cauchy-Schwarz inequality and Corollary 2.4, and further invokes the strong convexity of $\mathcal{L}(\theta, \alpha)$ in $\theta$. Combining Eq. (48) and Eq. (53), we derive the following upper bound on $[\mathcal{F}(\theta^t, \theta^*, \eta, \alpha^t)]^2 - [\mathcal{G}(\theta^t, \theta^*, \eta, \tilde{\alpha})]^2$:

$$\left[\mathcal{F}(\theta^t, \theta^*, \eta, \alpha^t)\right]^2 - \left[\mathcal{G}(\theta^t, \theta^*, \eta, \tilde{\alpha})\right]^2 \leq \mathcal{K}(\theta^t, \theta^*, \eta, \gamma, \tilde{\alpha}, \alpha^t), \tag{54}$$

$$\text{Where: } \mathcal{K}(\theta^t, \theta^*, \eta, \gamma, \tilde{\alpha}, \alpha^t) = -2\eta\left[\left(\frac{\mu^*}{2} - L^*\right)\|\theta^t - \theta^*\|^2 + \tilde{\alpha}^T(\mathcal{L}(\theta^t) - \mathcal{L}(\theta^*))\right.$$

$$\left. + \gamma\left(\frac{\mathcal{L}(\theta^{t-1})}{\mathbb{1}^T\mathcal{L}(\theta^{t-1})} - \tilde{\alpha}\right)^T(\mathcal{L}(\theta^t) - \mathcal{L}(\theta^*))\right]$$

$$+ 2\eta^2 L_\alpha L^* \gamma\left\|\frac{\mathcal{L}(\theta^{t-1})}{\mathbb{1}_k^T\mathcal{L}(\theta^{t-1})} - \tilde{\alpha}\right\|\|\theta^t - \theta^*\|. \tag{55}$$

Now, consider the following chain of inequalities deriving from Eq. (54):

$$\mathcal{K}(\theta^t, \theta^*, \eta, \gamma, \tilde{\alpha}, \alpha^t) \leq 0 \implies \left[\mathcal{F}(\theta^t, \theta^*, \eta, \alpha^t)\right]^2 - \left[\mathcal{G}(\theta^t, \theta^*, \eta, \tilde{\alpha})\right]^2 \leq 0$$
$$\implies \left[\mathcal{F}(\theta^t, \theta^*, \eta, \alpha^t)\right] \leq \left[\mathcal{G}(\theta^t, \theta^*, \eta, \tilde{\alpha})\right]. \tag{56}$$

Accordingly, we aim to find a condition on the mixing rate, $\gamma$, under which the chain of inequalities is satisfied. We proceed by letting $\mathcal{K}(\theta^t, \theta^*, \eta, \gamma, \tilde{\alpha}, \alpha^t) \leq 0$, and rearrange the terms:

$$\left(\frac{\mu^*}{2} - L^*\right)\|\theta^t - \theta^*\|^2 + \tilde{\alpha}^T(\mathcal{L}(\theta^t) - \mathcal{L}(\theta^*)) \geq \gamma\left[\eta L_\alpha L^*\left\|\frac{\mathcal{L}(\theta^{t-1})}{\mathbb{1}_k^T\mathcal{L}(\theta^{t-1})} - \tilde{\alpha}\right\|\|\theta^t - \theta^*\| \tag{57}\right.$$

$$\left. - \left(\frac{\mathcal{L}(\theta^{t-1})}{\mathbb{1}^T\mathcal{L}(\theta^{t-1})} - \tilde{\alpha}\right)^T(\mathcal{L}(\theta^t) - \mathcal{L}(\theta^*))\right].$$

We note that this chain of inequalities is satisfied if, for every $\eta > 0$, there exists a $\gamma$ such that:

$$\gamma \leq \frac{\left(\frac{\mu^*}{2} - L^*\right)\|\theta^t - \theta^*\|^2 + \tilde{\alpha}^T(\mathcal{L}(\theta^t) - \mathcal{L}(\theta^*))}{\eta L_\alpha L^*\left\|\frac{\mathcal{L}(\theta^{t-1})}{\mathbb{1}_k^T\mathcal{L}(\theta^{t-1})} - \tilde{\alpha}\right\|\|\theta^t - \theta^*\| - \left(\frac{\mathcal{L}(\theta^{t-1})}{\mathbb{1}^T\mathcal{L}(\theta^{t-1})} - \tilde{\alpha}\right)^T(\mathcal{L}(\theta^t) - \mathcal{L}(\theta^*))} \tag{58}$$

$$\leq \frac{\left(\frac{\mu^*}{2} - L^*\right)\|\theta^t - \theta^*\|^2 + \tilde{\alpha}^T(\mathcal{L}(\theta^t) - \mathcal{L}(\theta^*))}{\eta L_\alpha L^*\left\|\frac{\mathcal{L}(\theta^{t-1})}{\mathbb{1}_k^T\mathcal{L}(\theta^{t-1})} - \tilde{\alpha}\right\|\left[\|\theta^t - \theta^*\| - (\mathcal{L}(\theta^t) - \mathcal{L}(\theta^*))\right]} = \beta. \tag{59}$$

However, such a $\gamma$ exists iff the numerator and denominator in Eq. (59) have the same sign, ensuring that $\gamma > 0$. Accordingly, iff the condition provided in Eq. (60) is satisfied:

$$\left[\left(\frac{\mu^*}{2} - L^*\right)\|\theta^t - \theta^*\|^2 + \tilde{\alpha}^T(\mathcal{L}(\theta^t) - \mathcal{L}(\theta^*))\right]\left[\|\theta^t - \theta^*\| - (\mathcal{L}(\theta^t) - \mathcal{L}(\theta^*))\right] > 0, \tag{60}$$

It follows that for every learning rate $\eta > 0$ there exists a mixing rate $\gamma \in (0, \beta]$ satisfying Eq. (59) such that $\|(\theta^t - \eta\nabla_\theta\mathcal{L}(\theta^t, \alpha^t)) - \theta^*\| \leq \|(\theta^t - \eta\nabla_\theta\mathcal{L}(\theta^t, \tilde{\alpha})) - \theta^*\|$. $\qquad\square$

# B    DATASET DESCRIPTIONS

## B.1    MNIST DATASET

The **MNIST** (Modified National Institute of Standards and Technology) dataset is a collection of handwritten digits commonly used to train image processing systems. For the MNIST classification result from Section 4.1, the original training dataset, $J$, comprises $N = 60000$ samples, wherein the fixed-proportion mixing parameters (for default numerical class ordering of digits from $1 - 10$) are:

$$\tilde{\alpha} = [0.0987, 0.1124, 0.0993, 0.1022, 0.0974, 0.0904, 0.0986, 0.1044, 0.0975, 0.0991]^T$$

The test dataset, $K$, comprises $N_{\text{test}} = 10000$ samples, with class proportions equivalent to the class proportions in the base MNIST test dataset. For MNIST reconstruction (see Section 4.3), we utilize manual class imbalancing, reducing the number of samples comprising each numerical class $6 - 10$ by a factor of 5. The original training dataset, $J$, now contains $N = 36475$ samples, wherein the fixed-proportion mixing parameters (for default numerical class ordering of digits from $1 - 10$) are:

$$\tilde{\alpha} = [0.1624, 0.1848, 0.1633, 0.1681, 0.1602, 0.0297, 0.0324, 0.0344, 0.0321, 0.0326]^T$$

We note that the test dataset maintains the same class proportions as in the base MNIST test dataset. The features and labels within MNIST are summarized as follows:

- Each feature (image) is of size $28 \times 28$, representing grayscale intensities from 0 to 255.
- Target Variable: The numerical class (digit) the image represents, ranging from 1 to 10.

## B.2 FASHION MNIST DATASET

The **Fashion MNIST** dataset is a collection of clothing images commonly used to train image processing systems. For the Fashion MNIST classification result from Section 4.1, the original training dataset, $J$, consists of $N = 60000$ samples, wherein the fixed-proportion mixing parameters (for default numerical class ordering of clothing from $1 - 10$) are:

$$\tilde{\alpha} = [0.1, 0.1, 0.1, 0.1, 0.1, 0.1, 0.1, 0.1, 0.1, 0.1]^T = (0.1)\mathbb{1}_{10}$$

The test dataset, $K$, comprises $N_{\text{test}} = 10000$ samples, with class proportions equivalent to the class proportions in the base Fashion MNIST test dataset. For Fashion MNIST reconstruction (see Section 4.3), we use manual class imbalancing, reducing the number of samples within each numerical class $6 - 10$ by a factor of 5. The original training dataset $J$, now has $N = 36000$ samples. The fixed-proportion mixing parameters (for default numerical class ordering of clothing from $1 - 10$) are:

$$\tilde{\alpha} = [(0.1667)\mathbb{1}_5^T, (0.0333)\mathbb{1}_5^T]^T$$

We note that the test dataset maintains the same class proportions as in the base Fashion MNIST test dataset. The features and labels within Fashion MNIST are summarized as follows:

- Each feature (image) is of size $28 \times 28$, representing grayscale intensities from 0 to 255.
- Target Variable: The numerical class (clothing) the image represents, ranging from 1 to 10.

## B.3 CIFAR-10 DATASET

The **CIFAR-10** dataset is a collection of color images categorized into 10 different classes, and is commonly used to train image processing systems. For the CIFAR-10 classification result in Section 4.1, the original training dataset, $J$, comprises $N = 50000$ samples, wherein the fixed-proportion mixing parameters (for default numerical class ordering of categories from $1 - 10$) are:

$$\tilde{\alpha} = (0.1)\mathbb{1}_{10}$$

The test dataset, $K$, comprises $N_{\text{test}} = 10000$ samples, with class proportions equivalent to the class proportions in the base CIFAR-10 test dataset. For CIFAR-10 reconstruction (see Section 4.3), we use manual class imbalancing, reducing the number of samples in numerical classes $1 - 4, 7 - 10$ by a factor of 10. The original training dataset, $J$, now has $N = 14000$ samples. The fixed-proportion mixing parameters (for default numerical class ordering of categories from $1 - 10$) are:

$$\tilde{\alpha} = [(0.0357)\mathbb{1}_4^T, (0.3571)\mathbb{1}_2^T, (0.0357)\mathbb{1}_4^T]^T$$

We note that the test dataset maintains the same class proportions found in the base CIFAR-10 test dataset. The features and labels within CIFAR-10 are summarized as follows:

- Each feature (image) is of size $32 \times 32 \times 3$, with three color channels (RGB), and size 32 x 32 pixels for each channel, represented as a grayscale intensity from 0 to 255.
- Target Variable: The numerical class (category) the image represents, ranging from 1 to 10.

## B.4 IMAGENETTE DATASET

The **Imagenette** dataset contains a subset of 10 classes from the ImageNet dataset of color images, and is commonly used to train image processing systems. The base Imagenette training dataset, $I$, comprises $N_I = 9469$ samples, and the base Imagenette test dataset, $K$, comprises $N_{\text{test}} = 3925$ samples. For the Imagenette classification result in Section 4.1, we utilize manual class imbalancing. Let $N_i \in \mathbb{N}$ be the number of samples in each class, $i \in \{1, \ldots, 10\}$, from $I$, where $N_I = \sum_{i=1}^{10} N_i$. We define $\epsilon_i = 1 - 0.1i, \forall i \in \{1, \ldots, 10\}$ as the linearly decreasing *imbalance factor*. Accordingly, the original training dataset, $J$, has $N = \sum_{i=1}^{10} \epsilon_i N_i = 5207$ samples. The fixed-proportion mixing parameters (for default numerical class ordering of categories from $1 - 10$) are:

$$\tilde{\alpha} = [0.1849, 0.1650, 0.1525, 0.1152, 0.1083, 0.0918, 0.0737, 0.0536, 0.0365, 0.0184]^T$$

We note that the test dataset maintains the same class proportions found in the base Imagenette test dataset. The features and labels within Imagenette are summarized as follows:

- Each feature (image) is of size $224 \times 224 \times 3$, with three color channels (RGB), and size 224 x 224 pixels for each channel, represented as a grayscale intensity from 0 to 255.
- Target Variable: The numerical class (category) the image represents, ranging from 1 to 10.

## B.5 CIFAR-100 DATASET

The **CIFAR-100** dataset is a collection of color images categorized into 100 different classes, and is commonly used to train image processing systems. The base CIFAR-100 training dataset, $I$, has $N_I = 50000$ samples, and the base CIFAR-100 test dataset, $K$, has $N_{\text{test}} = 10000$ samples. For the CIFAR-100 classification result in Section 4.1, we utilize manual class imbalancing. Let $N_i \in \mathbb{N}$ be the number of samples in each class, $i \in \{1, \ldots, 100\}$, from $I$, whereby $N_I = \sum_{i=1}^{100} N_i$. We define $\epsilon_i = 40^{-i/100}, \forall i \in \{1, \ldots, 100\}$ as the logarithmically decreasing *imbalance factor*. Accordingly, the original training dataset, $J$, has $N = \sum_{i=1}^{100} \epsilon_i N_i = 13209$ samples. The fixed-proportion mixing parameters (for default numerical class ordering of categories from $1 - 100$) are:

$$\tilde{\alpha} = [\tilde{\alpha}_1, \tilde{\alpha}_2, \ldots, \tilde{\alpha}_{100}]^T, \quad \text{where:} \quad \tilde{\alpha}_i = (\epsilon_i N_i)/N, \ \forall i \in \{1, \ldots, 100\}$$

We note that the test dataset maintains the same class proportions found in the base CIFAR-100 test dataset. The features and labels within CIFAR-100 are summarized as follows:

- Each feature (image) is of size $32 \times 32 \times 3$, with three color channels (RGB), and size 32 x 32 pixels for each channel, represented as a grayscale intensity from 0 to 255.
- Target Variable: The numerical class (category) the image denotes, ranging from 1 to 100.

## B.6 IMDB DATASET

The **IMDB** dataset is a collection of movie reviews, categorized as positive or negative in sentiment. We split the IMDB dataset such that the base IMDB training dataset, $I$, has $N_I = 40000$ samples, and the base IMDB test dataset, $K$, consists of $N_{\text{test}} = 10000$ samples. For the IMDB classification result in Section 4.1, we leverage manual class imbalancing, wherein numerical class 1 retains $30\%$ of its samples. Accordingly, the original training dataset, $J$, has $N = 26000$ samples. The fixed-proportion mixing parameters (for default numerical class ordering of sentiment from $1, 2$) are:

$$\tilde{\alpha} = [0.2307, 0.7693]^T$$

We note that the test dataset maintains the same class proportions as in the base IMDB test dataset. The features and labels within the IMDB dataset are summarized as follows:

- Each feature (review) is tokenized and encoded as a sequence of word indices with a max length of 500 tokens. Sequences are padded or truncated to ensure uniform length.
- Target Variable: The numerical class (sentiment) the review represents, either 1 or 2.

## B.7 MEAN ESTIMATION DATASET

The **Mean Estimation** dataset is a synthetic benchmark designed for regression tasks, wherein each example, $(x_j, y_j)$, comprises a 10-dimensional feature vector, $x_j$, of samples from one of four statistical distributions, and the mean, $y_j$, of this distribution. We create an imbalanced original training dataset, $J$, with $N = 3000$ samples, where $J_1$ has 1000 examples drawn from a normal distribution with $\sigma = 1$, $J_2$ has 1000 examples drawn from an exponential distribution, $J_3$ has 800 examples drawn from a chi-squared distribution, and $J_4$ has 200 samples drawn from a uniform distribution. The fixed-proportion mixing parameters (for numerical ordering of distributions from $1 - 4$) are:

$$\tilde{\alpha} = [0.333, 0.333, 0.267, 0.067]^T$$

The test dataset, $K$, is created as a balanced dataset that has 1000 examples from each distribution, wherein $N_{\text{test}} = 4000$. The Mean Estimation dataset features and labels are summarized as follows:

- Each feature (vector of samples) is generated from one of four statistical distributions (normal, exponential, chi-squared, uniform). The feature vectors are created by sampling from these distributions with means uniformly drawn from the interval $[0, 1]$ for normal, exponential, and chi-squared distributions, and from $[20, 50]$ for the uniform distribution.
- Target Variable: The mean parameter used to generate the vector of samples, representing the underlying expected value of the chosen distribution.

### B.8 WINE QUALITY DATASET

The **Wine Quality** dataset consists of physicochemical tests on white and red wine samples, and the corresponding quality rating. We treat the wine type (white $= 1$, red $= 2$) as a categorical variable, wherein $k = 2$. We split the Wine Quality dataset such that the base Wine Quality training dataset, $J$, has $N = 3248$ samples, and the base Wine Quality test dataset, $K$, has $N_{\text{test}} = 3249$ samples. For the Wine Quality regression result in Section 4.2, we utilize manual class imbalancing, reducing the number of samples in numerical class 1 by a factor of 10. The original training dataset, $J$, now has $N = 1043$ samples, where the fixed-proportion mixing parameters (for numerical class ordering of wine type from $1, 2$) are:

$$\tilde{\alpha} = [0.234, 0.766]^T$$

We note that the test dataset maintains the same class proportions as in the base Wine Quality test dataset. The features and labels within the Wine Quality dataset are summarized as follows:

- Each feature (physicochemical tests) contains a set of test results, and is of size $11 \times 1$.

- Target Variable: The wine quality rating given to the set of physicochemical tests.

### B.9 CALIFORNIA HOUSING DATASET

The **California Housing** dataset contains housing data from California and their associated prices. As the ocean proximity variable is categorical ($<$1H OCEAN $= 1$, INLAND $= 2$, NEAR BAY $= 3$, NEAR OCEAN $= 4$), we denote $k = 4$. We split the California Housing dataset such that the base California Housing training dataset, $J$, has $N = 10214$ samples, and the base California Housing test dataset, $K$, has $N_{\text{test}} = 10214$ samples. For the California Housing regression result in Section 4.2, we use manual class imbalancing, reducing the number of samples in numerical classes $1, 2, 4$ by a factor of 20. The original training dataset, $J$, now has $N = 3641$ samples. The fixed-proportion mixing parameters (for numerical class ordering of ocean proximity from $1 - 4$) are:

$$\tilde{\alpha} = [0.0615, 0.9055, 0.0154, 0.0176]^T$$

We note that the test dataset maintains the same class proportions as in the base California Housing test dataset. The features and labels in the California Housing dataset are summarized as follows:

- Each feature (housing data) contains various housing attributes, and is of size $8 \times 1$.

- Target Variable: The housing price associated with the housing data.

## C EXPERIMENT DETAILS

### C.1 NEURAL NETWORK ARCHITECTURES

We provide comprehensive descriptions for six different neural network architectures designed for various tasks: classification, regression, and image reconstruction. Each of these architectures were employed to generate the respective empirical results pertaining to the aforementioned tasks.

#### C.1.1 FULLY CONNECTED NETWORKS

We leverage fully connected networks in our analysis for regression on Mean Estimation, California Housing, and Wine Quality. The network consists of the following layers, wherein $d = 10$ for Mean Estimation, $d = 11$ for Wine Quality, and $d = 8$ for California Housing:

- **Fully Connected Layer (`fc1`)**: Transforms the input features from a $d$-dimensional space to a 64-dimensional space.

- **ReLU Activation (`relu`)**: Applies the ReLU activation function to the output of `fc1`.

- **Fully Connected Layer (`fc2`)**: Maps the 64-dimensional representation from `relu` to a 1-dimensional output.

### C.1.2 CONVOLUTIONAL NEURAL NETWORKS

We utilize the LeNet-5 convolutional neural network architecture in our analysis for image classification on MNIST and Fashion MNIST. The network consists of the following layers:

- **Convolutional Layer (`conv1`)**: Applies a 2D convolution with 1 input channel, 6 output channels, and a kernel size of 5.
- **ReLU Activation (`relu1`)**: Applies the ReLU activation function to the output of `conv1`.
- **Max Pooling Layer (`pool1`)**: Performs 2x2 max pooling on the output of `relu1`.
- **Convolutional Layer (`conv2`)**: Applies a 2D convolution with 6 input channels, 16 output channels, and a kernel size of 5.
- **ReLU Activation (`relu2`)**: Applies the ReLU activation function to the output of `conv2`.
- **Max Pooling Layer (`pool2`)**: Performs 2x2 max pooling on the output of `relu2`.
- **Flatten Layer**: Reshapes the pooled feature maps into a 1D vector.
- **Fully Connected Layer (`fc1`)**: Maps the flattened vector to a 120-dimensional space.
- **ReLU Activation (`relu3`)**: Applies the ReLU activation function to the output of `fc1`.
- **Fully Connected Layer (`fc2`)**: Maps the 120-dimensional input to a 84-dimensional space
- **ReLU Activation (`relu4`)**: Applies the ReLU activation function to the output of `fc2`.
- **Fully Connected Layer (`fc3`)**: Produces a 10-dimensional output for classification.

For image classification on CIFAR-10 and CIFAR-100, we employ an adapted, larger version of the LeNet-5 model. The network consists of the following layers, wherein $k = 10$ for CIFAR-10 and $k = 100$ for CIFAR-100.

- **Convolutional Layer (`conv1`)**: Applies 2D convolution with 3 input channels, 16 output channels, and a kernel size of 3.
- **ReLU Activation (`relu1`)**: Applies the ReLU activation function to the output of `conv1`.
- **Max Pooling Layer (`pool1`)**: Performs 2x2 max pooling on the output of `relu1`.
- **Convolutional Layer (`conv2`)**: Applies 2D convolution with 16 input channels, 32 output channels, and a kernel size of 3.
- **ReLU Activation (`relu2`)**: Applies the ReLU activation function to the output of `conv2`.
- **Max Pooling Layer (`pool2`)**: Performs 2x2 max pooling on the output of `relu2`.
- **Convolutional Layer (`conv3`)**: Applies 2D convolution with 32 input channels, 64 output channels, and a kernel size of 3.
- **ReLU Activation (`relu3`)**: Applies the ReLU activation function to the output of `conv3`.
- **Max Pooling Layer (`pool3`)**: Performs 2x2 max pooling on the output of `relu3`.
- **Flatten Layer**: Reshapes the pooled feature maps into a 1D vector of size $4 \times 4 \times 64$.
- **Fully Connected Layer (`fc1`)**: Maps the flattened vector to a 500-dimensional space.
- **ReLU Activation (`relu4`)**: Applies the ReLU activation function to the output of `fc1`.
- **Dropout Layer (`dropout1`)**: Applies dropout with $p = 0.5$ to the output of `relu4`.
- **Fully Connected Layer (`fc2`)**: Produces a $k$-dimensional output for classification.

### C.1.3 RESIDUAL NEURAL NETWORKS

For image classification on Imagenette, we employ the ResNet-18 residual neural network architecture, which consists of the following layers:

- **Convolutional Layer (`conv1`)**: Applies a 7x7 convolution with 3 input channels, 64 output channels, and a stride of 2.
- **Batch Normalization (`bn1`)**: Normalizes the output of `conv1`.
- **ReLU Activation (`relu`)**: Applies the ReLU activation function to the output of `bn1`.

- **Max Pooling Layer (`maxpool`)**: Performs 3x3 max pooling with a stride of 2 on the output of `relu`.

- **Residual Layer 1 (`layer1`)**: Contains two residual blocks, each with 64 channels.

- **Residual Layer 2 (`layer2`)**: Contains two residual blocks, each with 128 channels.

- **Residual Layer 3 (`layer3`)**: Contains two residual blocks, each with 256 channels.

- **Residual Layer 4 (`layer4`)**: Contains two residual blocks, each with 512 channels.

- **Average Pooling (`avgpool`)**: Applies adaptive average pooling to reduce the spatial dimensions to 1x1.

- **Fully Connected Layer (`fc`)**: Produces a 10-dimensional output for classification.

### C.1.4 TRANSFORMER MODELS

For sentiment classification on IMDB Sentiment Analysis, we leverage a transformer architecture, which consists of the following layers:

- **Embedding Layer (`embedding`)**: Maps input tokens to 64-dimensional embeddings.

- **Positional Encoding (`pos_encoder`)**: Adds positional information to the embeddings with a maximum sequence length of 500.

- **Transformer Encoder (`transformer_encoder`)**: Applies a transformer encoder with 1 layer, 4 attention heads, and a hidden dimension of 128.

- **Pooling Layer (`pool`)**: Averages the transformer outputs across the sequence length.

- **Dropout Layer (`dropout`)**: Applies dropout with probability 0.1 to the pooled output.

- **Fully Connected Layer (`fc1`)**: Maps the 64-dimensional pooled vector to 32-dimensional space.

- **ReLU Activation (`relu1`)**: Applies the ReLU activation function to the output of `fc1`.

- **Fully Connected Layer (`fc2`)**: Maps the 32-dimensional input to 2 output classes.

### C.1.5 AUTOENCODER MODELS

For image reconstruction on MNIST, Fashion MNIST, and CIFAR-10, we employ an autoencoder. This network consists of the following layers, where $d = 784$ for MNIST and Fashion MNIST, and $d = 3072$ for CIFAR-10:

- **Fully Connected Layer (`fc1`)**: Transforms the input features from a $d$-dimensional space to a 128-dimensional space.

- **ReLU Activation (`relu1`)**: Applies the ReLU activation function to the output of `fc1`.

- **Fully Connected Layer (`fc2`)**: Reduces the 128-dimensional representation to a 32-dimensional encoded vector.

- **Fully Connected Layer (`fc3`)**: Expands the 32-dimensional encoded vector back to a 128-dimensional space.

- **ReLU Activation (`relu1`)**: Applies the ReLU activation function to the output of `fc3`.

- **Fully Connected Layer (`fc4`)**: Maps the 128-dimensional representation back to the original $d$-dimensional space.

- **Sigmoid Activation (`sigmoid1`)**: Applies the Sigmoid activation function to ensure the output values are between 0 and 1.

### C.2 FOCAL TRAINING

For the classification tasks outlined in Section 4.1, we compare learn2mix and classical training with focal loss-based neural network training (focal training). Let $\tilde{\alpha} \in [0, 1]^k$ denote the vector of fixed-proportion mixing parameters, let $\mathcal{L}(\theta^t) \in \mathbb{R}^k$ denote the vector of class-wise cross entropy losses

at time $t$, and let $\omega \in \mathbb{R}^k$ denote the vector of class-wise weighting factors, where $\forall i \in \{1, \ldots, k\}$:

$$\omega_i = \frac{[1/(\tilde{\alpha}_i N)]}{\sum_{i'=1}^{k} [1/(\tilde{\alpha}_{i'} N)]} \times k. \tag{61}$$

The vector of predicted class-wise probabilities, $p \in [0, 1]^k$, is given by $p = \exp\left(-\mathcal{L}(\theta^t)\right)$, and we let $\Gamma \in \mathbb{R}_{\geq 0}$ be the focusing parameter. The focal loss at time $t$, $\mathcal{L}_{\text{FCL}}(\theta^t, \omega) \in \mathbb{R}_{\geq 0}$, is given by:

$$\mathcal{L}_{\text{FCL}}(\theta^t, \tilde{\alpha}) = \frac{1}{k} \sum_{i=1}^{k} (-\omega_i)(1 - p_i)^{\Gamma} \log(p_i). \tag{62}$$

Per the recommendations in (Lin et al., 2017), we choose $\Gamma = 2$ in compiling the empirical results.

## C.3 SMOTE TRAINING

For the classification tasks outlined in Section 4.1, we also compare learn2mix and classical training with neural networks trained on SMOTE-oversampled datasets (SMOTE training). Let $J$ denote the original training dataset, where the number of samples in each class, $i \in \{1, \ldots, k\}$ is given by $\tilde{\alpha}_i N$. After applying SMOTE oversampling, we obtain a new training dataset, $J^{\text{SMOTE}}$, with uniform class proportions, $\tilde{\alpha}_i^{\text{SMOTE}} = \frac{1}{k}, \forall i \in \{1, \ldots, k\}$. The total number of samples in $J^{\text{SMOTE}}$, is given by:

$$N^{\text{SMOTE}} = \left( \max_{i \in \{1, \ldots, k\}} \tilde{\alpha}_i N \right) \times k. \tag{63}$$

In the original training dataset, $J$, we use a batch size of $M$, resulting in $P = \frac{N}{M}$ total batches. For consistency with learn2mix and classical training (see Section 4.1), we perform SMOTE training on $P$ batches of size $M$ from the SMOTE oversampled training dataset, $J^{\text{SMOTE}}$, during each epoch.

## C.4 NEURAL NETWORK TRAINING HYPERPARAMETERS

The relevant hyperparameters used to train the neural networks outlined in Section C.1 are provided in Table 3. All results presented in the main text were produced using these hyperparameter choices.

Table 3: Neural network training hyperparameters (grouped by task).

| Dataset | Task | Optimizer | Learning Rate ($\eta$) | Mixing Rate ($\gamma$) (Learn2Mix) | Batch Size ($M$) |
|---------|------|-----------|------------------------|-------------------------------------|------------------|
| MNIST | Classification | Adam | 1.0e-5 | 0.1 | 1000 |
| Fashion MNIST | Classification | Adam | 5.0e-6 | 0.5 | 1000 |
| CIFAR-10 | Classification | Adam | 1.0e-5 | 0.1 | 1000 |
| Imagenette | Classification | Adam | 1.0e-6 | 0.1 | 100 |
| CIFAR-100 | Classification | Adam | 0.0001 | 0.5 | 5000 |
| IMDB | Classification | Adam | 0.0001 | 0.1 | 500 |
| Mean Estimation | Regression | Adam | 5.0e-5 | 0.01 | 500 |
| Wine Quality | Regression | Adam | 0.0001 | 0.05 | 100 |
| California Housing | Regression | Adam | 5.0e-5 | 0.01 | 1000 |
| MNIST | Reconstruction | Adam | 0.0005 | 0.1 | 1000 |
| Fashion MNIST | Reconstruction | Adam | 1.0e-5 | 0.1 | 1000 |
| CIFAR-10 | Reconstruction | Adam | 1.0e-5 | 0.1 | 1000 |

