# OpenReview forum: "Learn2Mix: Training Neural Networks Using Adaptive Data Integration"
_ICLR.cc/2025/Conference — Submitted to ICLR 2025_

### Official Review · Reviewer_k322 · 2024-11-01

**Soundness:** 2
**Presentation:** 3
**Contribution:** 2
**Rating:** 5
**Confidence:** 3

**Summary:**

This paper introduces "learn2mix", a training strategy for neural networks that dynamically adjust the weight of each class during training according to the class-wise error, which is different from the traditional methods that use a static class proportion. Using a dynamic class proportion accelerate the converge rate and improve the performance, especially when the class is imbalanced.

**Strengths:**

1. The paper is well-written and easy to follow. The topic is related to class-imbalance and neural network training which matches ICLR well. The proposed approach is simple and effective.

2. The paper provides both theoritical and emperical justification.

**Weaknesses:**

1. My biggest concern about this paper is the evaluation. The only baseline that the authors compared is the standard training mechanisms, although curriculum learning has already been widely studied in the literature. The contribution would be much more convincing by having stronger baselines, such as using Focal loss or any curriculum learning methods.

2. The results shown in Figures 1 and 2 don't seem to be fully converged.

**Questions:**

1. How does the proposed method perform compared with stronger baselines for class imbalance, such as using the Focal loss?

2. How should a user choose the mixing rate? Is the proposed method sensitive to the hyper-parameter?

3. Since the proposed method doesn't use a fixed class weight. I wondered if this approach is more robust to distributional shifts regarding the class imbalance levels.

**Details Of Ethics Concerns:**

No ethics concerns

---

> ### Author Response · Authors · 2024-11-20
> **Rebuttal by Authors**
>
> Dear Reviewer k322,
>
> Thank you for your helpful feedback. Below, we have provided responses to your comments.
>
> > My biggest concern about this paper is the evaluation. The only baseline that the authors compared is the standard training mechanisms, although curriculum learning has already been widely studied in the literature. The contribution would be much more convincing by having stronger baselines, such as using Focal loss or any curriculum learning methods.
>
> We appreciate your feedback regarding the evaluation of Learn2Mix. In response to your concern and similar suggestions from Reviewer Cg5R, we have uploaded a new version of our paper that includes comparisons with Focal Loss [1] and SMOTE [2] (a widely used oversampling technique that addresses class imbalance). While both focal loss-based training and SMOTE-based training yield improved performance over classical training across nearly every classification benchmark, our empirical results demonstrate that Learn2Mix maintains faster convergence and improved classification accuracy over focal loss-based training and SMOTE-based training. We have provided the complete code in the supplementary materials to reproduce all the results in the main text.
>
> > The results shown in Figures 1 and 2 don't seem to be fully converged.
>
> In response to your feedback, we have uploaded a new version of our paper in which we have updated the models and increased the training time to better depict the accelerated convergence behavior afforded by Learn2Mix in classification tasks. We want to note that, as emphasized in our abstract and in the introduction, Learn2Mix is intended for applications in _resource-constrained environments_, where computational resources are limited and extensive training periods using larger models are not feasible. Our focus is on showcasing the accelerated learning and performance improvements that Learn2Mix offers within these practical constraints, illustrating that it achieves superior performance compared to classical training methods in shorter time frames.
>
> > How does the proposed method perform compared with stronger baselines for class imbalance, such as using the Focal loss?
>
> As we have mentioned above, we have updated our paper to include comparisons with Focal Loss [1] and SMOTE [2]. Across all tested classification tasks, Learn2Mix consistently demonstrates accelerated convergence and improved performance compared to both focal loss-based training and SMOTE-based training.
>
> > How should a user choose the mixing rate? Is the proposed method sensitive to the hyper-parameter?
>
> Thank you for raising this question. In the updated version of the paper, we have added a comment in the empirical results section addressing this concern. Practically, we observe that under Learn2Mix training, selecting a mixing rate, $\gamma$, between 0.01 and 0.5 yields the best performance across various datasets and tasks. This suggests that users can choose a mixing rate within this range without requiring extensive hyperparameter tuning.
>
> > Since the proposed method doesn't use a fixed class weight. I wondered if this approach is more robust to distributional shifts regarding the class imbalance levels.
>
> Since our method does not rely on fixed class weights, it is indeed more adaptable to changes in class distributions. Our empirical studies include datasets with varying degrees of class imbalance—such as Imagenette, CIFAR-100, and IMDB for classification—and demonstrate that Learn2Mix consistently achieves accelerated convergence and improved performance. Furthermore, even in balanced datasets like MNIST, Fashion MNIST, and CIFAR-10, Learn2Mix outperforms classical training, focal loss-based training, and SMOTE-based training. This indicates that Learn2Mix is robust to distributional shifts regarding class imbalance levels, effectively handling both imbalanced and balanced scenarios.
>
> Thank you again for your constructive feedback, which has been significant in helping enhance our paper.
>
> **References:**
>
> [1] Tsung-Yi Lin, Priya Goyal, Ross Girshick, Kaiming He, and Piotr Dollar. Focal loss for dense object detection. In Proceedings of the IEEE International Conference on Computer Vision, pp. 2980–2988, 2017.
>
> [2] Nitesh V Chawla, Kevin W Bowyer, Lawrence O Hall, and W Philip Kegelmeyer. SMOTE: Synthetic minority over-sampling technique. In Proceedings of the 2002 Joint Conference on IEEE International Conference on Knowledge Discovery and Data Mining and IEEE European Conference on Machine Learning, pp. 878–884. IEEE, 2002.

---

> > ### Author Response · Authors · 2024-11-25
> > **Kind Reminder**
> >
> > Dear Reviewer k322,
> >
> > As we near the end of this discussion period, we wish to extend our sincere gratitude for the time and effort you have invested in reviewing our work.
> >
> > This message serves as a gentle reminder to kindly let us know if you have any further questions we can assist with and if you are considering adjusting your assessment of our work based on the feedback received.
> >
> > Best regards,
> > The Authors

---

> > > ### Comment · Reviewer_k322 · 2024-11-26
> > >
> > > Thank you authors for improving the maniscript and conducing extra experiments. It's interesting to see that the updated experiments with Focal Loss. I increased my score accordingly.

---

### Official Review · Reviewer_Cg5R · 2024-11-02

**Soundness:** 2
**Presentation:** 3
**Contribution:** 2
**Rating:** 5
**Confidence:** 4

**Summary:**

The authors present learn2mix, a training method which dynamically adapts the proportion of classes during training using class-wise error rates. They provide theoretical justifications for the performance of learn2mix and also empirically demonstrate accelerated convergence on balanced and imbalanced datasets.

**Strengths:**

- The method is presented in a clear manner and ensures each example within the class-specific dataset will be chosen uniformly at random through training, even with the reweighting.
- The experiment details are presented clearly between the main text and the appendix, and each experiment seems replicable.
- learn2mix outperforms classical training across various classification and regression tasks.

**Weaknesses:**

- The empirical results for both classical and learn2mix are concerningly underwhelming. For example, the performance of both methods are cut off at only <40% accuracy on CIFAR-10, even though most image classification models can easily achieve 90% accuracy, with modern models reporting >99% test accuracy. Even when considering the LeNet-5 architecture which the authors benchmarked, the original LeNet-5 and MNIST paper [1] achieves 99% accuracy in 20 epochs, while this paper reports <80% accuracy in 60 epochs. This significant gap in performance makes it unclear if the faster convergence of learn2mix holds in practice with more careful training setups and realistic models.
- The method was only compared against classical empirical risk minimization. Classical training is known to be ineffective for class imbalance, so it would be helpful to see how this method compares to other methods which adjust the training data such as oversampling.
- The theoretical results require that the class-wise loss is strongly convex in $\theta$; however, the loss landscape for neural networks is known to be highly non-convex [2], which challenges the relevance of their findings.

[1] LeCun et al, Gradient Based Learning Applied to Document Recognition

[2] Li et al, Visualizing the Loss Landscape of Neural Nets

**Questions:**

- How does learn2mix, which modifies the composition of the training data, compare with other regimes such as [3], which modify the loss function? This method seems slightly more expensive because it involves the extra step of dynamically composing the training data.

[3] Sagawa et al, Distributionally Robust Neural Networks for Group Shifts: On the Importance of Regularization for Worst-Case Generalization

---

> ### Author Response · Authors · 2024-11-20
> **Rebuttal by Authors**
>
> Dear Reviewer Cg5R,
>
> Thank you for your detailed feedback. Below, we have provided responses to your comments.
>
> > The empirical results for both classical and learn2mix are concerningly underwhelming. For example, the performance of both methods are cut off at only <40% accuracy on CIFAR-10, even though most image classification models can easily achieve 90% accuracy, with modern models reporting >99% test accuracy. Even when considering the LeNet-5 architecture which the authors benchmarked, the original LeNet-5 and MNIST paper [1] achieves 99% accuracy in 20 epochs, while this paper reports <80% accuracy in 60 epochs. This significant gap in performance makes it unclear if the faster convergence of learn2mix holds in practice with more careful training setups and realistic models.
>
> As stated in our abstract and emphasized in the introduction, Learn2Mix is intended for applications in _resource-constrained environments_, where computational resources are limited, and extensive training periods using larger models are not feasible. Despite these constraints, our paper also includes more realistic models in our experiments, including ResNet-based architectures on the Imagenette dataset and Transformer-based architectures on the IMDB dataset, to demonstrate the applicability of Learn2Mix in practical settings with larger models. In response to your concerns, we have also uploaded a new version of the paper with updated empirical results, achieving over 90% accuracy on MNIST with LeNet-5, and over 60% accuracy on CIFAR-10 with LeNet-5 (which is known to achieve between 60-70% accuracy on CIFAR-10). The difference in performance compared to the original LeNet-5 paper is due to our use of a lower learning rate to simulate more careful training setups aimed at improving model convergence and generalization. Across the updated training settings, Learn2Mix depicts accelerated convergence over classical training and focal loss-based training [1].
>
> > The method was only compared against classical empirical risk minimization. Classical training is known to be ineffective for class imbalance, so it would be helpful to see how this method compares to other methods which adjust the training data such as oversampling.
>
> Thank you for your valuable suggestion. In response to similar feedback from Reviewer k322, we have updated our paper with new empirical results that include comparisons with focal loss-based training [1], which is recognized for achieving state-of-the-art performance in various class imbalance settings. While focal loss-based training shows improved performance over classical methods on datasets like MNIST and Fashion MNIST, our results demonstrate that Learn2Mix still achieves faster convergence. We have provided the complete code in the supplementary materials to reproduce all the results in the main text. Moreover, it is important to note that methods such as oversampling and class-balanced loss functions, including focal loss, are predominantly applicable to classification tasks. In contrast, Learn2Mix extends beyond classification and is also applicable to regression and reconstruction settings. Our empirical findings confirm that Learn2Mix accelerates convergence across various regression and reconstruction tasks, highlighting its broad applicability.
>
> > The theoretical results require that the class-wise loss is strongly convex in $\theta$; however, the loss landscape for neural networks is known to be highly non-convex, which challenges the relevance of their findings.
>
> We note that assumptions such as strong convexity are standard in the machine learning literature when deriving convergence rates for gradient-based optimization methods [2], [3]. These assumptions provide a manageable framework to obtain rigorous and provable insights into the behavior of optimization algorithms. Without them, establishing theoretical convergence guarantees becomes exceedingly challenging. While the strong convexity assumption may not hold strictly in practical neural network training, our theoretical findings offer valuable intuition about the convergence properties of Learn2Mix. Moreover, our empirical results demonstrate that Learn2Mix effectively accelerates convergence in real-world non-convex settings, reinforcing the practical relevance of our work.
>
> (rebuttal continued in comment)

---

> > ### Author Response · Authors · 2024-11-20
> > **Continuation of Rebuttal**
> >
> > > How does learn2mix, which modifies the composition of the training data, compare with other regimes which modify the loss function? This method seems slightly more expensive because it involves the extra step of dynamically composing the training data.
> >
> > We have uploaded a new version of our paper where we benchmark Learn2Mix against Focal Loss [1], which adjusts the loss function to focus on harder, misclassified examples. Across all tested classification tasks, Learn2Mix consistently demonstrates accelerated convergence compared to both focal loss-based training and classical training methods. Regarding computational overhead, the primary difference with Learn2Mix is in batch formation: while classical training shuffles the entire dataset at the start of each epoch and then forms batches, Learn2Mix shuffles samples within each class and selects subsets based on mixing parameters to form batches. This process introduces minimal additional computational complexity relative to the forward and backward passes of the neural network. The overhead of storing and updating the $k$ mixing parameters (in a $k$-class classification problem) is also negligible compared to the total number of parameters in the neural network (e.g., ResNet-18, which has 17 million parameters).
> >
> > Thank you again for your valuable feedback, which has been important in helping improve our paper.
> >
> > **References:**
> >
> > [1] Tsung-Yi Lin, Priya Goyal, Ross Girshick, Kaiming He, and Piotr Dollar. Focal loss for dense object detection. In Proceedings of the IEEE International Conference on Computer Vision, pp. 2980–2988, 2017.
> >
> > [2] Guanghui Wang, Shiyin Lu, Quan Cheng, Wei wei Tu, and Lijun Zhang. SAdam: A variant of adam for strongly convex functions. In International Conference on Learning Representations, 2020.
> >
> > [3] Arindam Banerjee, Pedro Cisneros-Velarde, Libin Zhu, and Misha Belkin. Restricted Strong Convexity of Deep Learning Models with Smooth Activations. In International Conference on Learning Representations, 2023.

---

> > > ### Comment · Reviewer_Cg5R · 2024-11-20
> > >
> > > Thank you for your effort in the rebuttal, and I'm glad to see updated experiments with improved accuracy!
> > >
> > > For the new focal-loss experiments, I'm surprised that they underperform compared to classical training in every single data-imbalanced setting, especially when this loss was specifically designed for these scenarios. Furthermore, I'm noting that the train accuracy even decreases for focal loss on Imagenette. Do you have any intuitions for this behavior? If this is due to the extreme level of class imbalance, it would also be interesting to see the behavior of these models in a less imbalanced setting (maybe 1:2 or 1:5 rather than 1:20).
> > >
> > > Furthermore, while focal-loss is a commonly used class-imbalance method, I think the most relevant work might be static dataset resampling. In your introduction, you state that "While existing approaches address class imbalance by adjusting sample weights or data resampling, they do not dynamically change the class-wise composition... In contrast with classical training schemes that have fixed class proportions... [learn2mix's] dynamic adjustment facilitates faster convergence and improved performance." However, I do not see evidence of this claim in the current manuscript, since there are no comparison between the dynamic weighting that this paper proposes with static weighting methods.

---

> ### Author Response · Authors · 2024-11-22
> **Response to Reviewer**
>
> Dear Reviewer Cg5R,
>
> Thank you for your follow-up comments. Below, we have addressed your questions.
>
> > For the new focal-loss experiments, I'm surprised that they underperform compared to classical training in every single data-imbalanced setting, especially when this loss was specifically designed for these scenarios. Furthermore, I'm noting that the train accuracy even decreases for focal loss on Imagenette. Do you have any intuitions for this behavior? If this is due to the extreme level of class imbalance, it would also be interesting to see the behavior of these models in a less imbalanced setting (maybe 1:2 or 1:5 rather than 1:20).
>
> Regarding the performance of focal loss, this behavior is indeed due to the extreme level of class imbalance in our considered versions of Imagenette and CIFAR-100. To note, we chose this ratio to specifically demonstrate the robustness of learn2mix to extreme levels of class imbalance, which is a regime where focal loss falters. In response to your feedback, we have uploaded a new version of the paper with updated experiments on CIFAR-100 and Imagenette to evaluate less extreme class imbalance patterns (see Section 4.1 of the main text). In particular, we now use a logarithmically decreasing ratio for CIFAR-100 (detailed in Section B.5 of the Appendix) and a linearly decreasing ratio for Imagenette (detailed in Section B.4 of the Appendix). In this updated regime, we observe that focal loss now outperforms classical training on both Imagenette and CIFAR-100. However, learn2mix maintains faster convergence and achieves improved performance over classical training, focal loss-based training, and SMOTE-based training [4].
>
> > Furthermore, while focal-loss is a commonly used class-imbalance method, I think the most relevant work might be static dataset resampling. In your introduction, you state that "While existing approaches address class imbalance by adjusting sample weights or data resampling, they do not dynamically change the class-wise composition... In contrast with classical training schemes that have fixed class proportions... [learn2mix's] dynamic adjustment facilitates faster convergence and improved performance." However, I do not see evidence of this claim in the current manuscript, since there are no comparison between the dynamic weighting that this paper proposes with static weighting methods.
>
> Thank you for your suggestion. In the updated version of the paper, we have further benchmarked the performance of learn2mix versus SMOTE-based training [4], a widely used static dataset resampling method that produces a new training dataset by oversampling the minority classes (detailed in Section C.3 of the Appendix). Across all tested classification benchmarks (see Section 4.1 of the main text), we observe that learn2mix maintains accelerated convergence and improved performance over SMOTE-based training, focal loss-based training, and classical training. We have provided the complete code in the supplementary materials to reproduce all the results in the main text.
>
> **References:**
>
> [4] Nitesh V Chawla, Kevin W Bowyer, Lawrence O Hall, and W Philip Kegelmeyer. SMOTE: Synthetic minority over-sampling technique. In Proceedings of the 2002 Joint Conference on IEEE International Conference on Knowledge Discovery and Data Mining and IEEE European Conference on Machine Learning, pp. 878–884. IEEE, 2002.

---

> > ### Author Response · Authors · 2024-11-25
> > **Kind Reminder**
> >
> > Dear Reviewer Cg5R,
> >
> > As we near the end of this discussion period, we wish to extend our sincere gratitude for the time and effort you have invested in reviewing our work.
> >
> > This message serves as a gentle reminder to kindly let us know if you have any further questions we can assist with and if you are considering adjusting your assessment of our work based on the feedback received.
> >
> > Best regards,
> > The Authors

---

> > > ### Comment · Reviewer_Cg5R · 2024-12-02
> > >
> > > Thank you for your response! It's encouraging to see the performance of Learn2Mix when compared with other class-imbalance methods, and I have updated my score to reflect the new experiments.

---

### Official Review · Reviewer_5wxS · 2024-11-04

**Soundness:** 3
**Presentation:** 3
**Contribution:** 3
**Rating:** 6
**Confidence:** 3

**Summary:**

This paper addresses the limitation of classical training paradigms that maintain fixed class proportions within batches, which fails to account for varying levels of difficulty across different classes and can hinder optimal convergence rates. The authors propose Learn2Mix, a training strategy that dynamically adjusts the proportion of classes within batches based on real-time class-wise error rates, directing more training emphasis towards challenging or underperforming classes, thereby accelerating model convergence and improving performance across classification, regression, and reconstruction tasks.

**Strengths:**

1. The paper is well-structured and highly readable. Its motivation stems from a key observation: different classes have varying learning difficulties, an aspect that traditional class imbalance algorithms have largely overlooked.
3. The paper proposes a novel approach where class proportions are dynamically adjusted based on training loss. The idea is both innovative and intuitive.
3. The experimental validation is comprehensive and convincing, encompassing six datasets including CIFAR-100 with its 100 classes, and spanning three different tasks: classification, regression, and image reconstruction.
4. While I haven't thoroughly examined the convergence analysis proofs, the paper provides theoretical guarantees for its approach.

**Weaknesses:**

The class imbalance setup for CIFAR-100 appears oversimplified, where the first 50 classes each comprise 0.1% of the data while the latter 50 classes each comprise 1.9%. A more rigorous evaluation should explore diverse imbalance patterns, such as exponentially decreasing ratios, step-wise distributions, or even real-world imbalance scenarios, to better validate the algorithm's robustness under complex class distributions.

**Questions:**

Could the evaluation be strengthened by testing more complex class imbalance patterns on CIFAR-100, rather than simply assigning 0.1% to the first 50 classes and 1.9% to the remaining ones?

---

> ### Author Response · Authors · 2024-11-20
> **Rebuttal by Authors**
>
> Dear Reviewer 5wxS,
>
> Thank you for your valuable feedback. Below, we have addressed your comments.
>
> > The class imbalance setup for CIFAR-100 appears oversimplified, where the first 50 classes each comprise 0.1% of the data while the latter 50 classes each comprise 1.9%. A more rigorous evaluation should explore diverse imbalance patterns, such as exponentially decreasing ratios, step-wise distributions, or even real-world imbalance scenarios, to better validate the algorithm's robustness under complex class distributions.
>
> Assessing the robustness of Learn2Mix in diverse imbalance scenarios is an important aspect of our ongoing research. Apart from CIFAR-100, we have provided several datasets with less uneven class distributions across classification, regression, and reconstruction tasks, such as IMDB for classification, and Mean Estimation and Wine Quality for regression. Furthermore, our empirical results demonstrate that even in balanced class distribution settings (i.e., MNIST, Fashion MNIST, and CIFAR-10), Learn2Mix converges faster than both classical training and focal loss-based training [1] (added to the new uploaded version of our paper). Overall, the trend we observe is that Learn2Mix achieves better performance with greater class imbalance, but also maintains improved performance in scenarios without class imbalance.
>
> > Could the evaluation be strengthened by testing more complex class imbalance patterns on CIFAR-100, rather than simply assigning 0.1% to the first 50 classes and 1.9% to the remaining ones?
>
> Our study aims to demonstrate the versatility of Learn2Mix across a spectrum of imbalance scenarios: from datasets without class imbalance (i.e., MNIST, Fashion MNIST, and CIFAR-10), through those with moderate imbalance (i.e., IMDB, Mean Estimation, and Wine Quality), to ones with significant imbalance (e.g., CIFAR-100). The specific imbalance setup in CIFAR-100—assigning 0.1% to the first 50 classes and 1.9% to the remaining—was intentionally designed to highlight Learn2Mix's robustness in extreme conditions. Exploring more complex class imbalance patterns is an important aspect of our ongoing work.
>
> Thank you again for your valuable feedback, which has been instrumental in helping improve our paper.
>
> **References:**
>
> [1] Tsung-Yi Lin, Priya Goyal, Ross Girshick, Kaiming He, and Piotr Dollar. Focal loss for dense object detection. In Proceedings of the IEEE International Conference on Computer Vision, pp. 2980–2988, 2017.

---

> > ### Comment · Reviewer_5wxS · 2024-11-21
> > **Continuation of Rebuttal**
> >
> > Thanks for the reply. I know you have tested your method on a series of experiments on datasets except from cifar100. However, what I am interested in is the performance on a more difficult cifar100 case as I explained before.

---

> ### Author Response · Authors · 2024-11-22
> **Response to Reviewer**
>
> Dear Reviewer 5wxS,
>
> Thank you for your follow-up comment. In response to your feedback, we have uploaded a new version of the paper with updated experiments on CIFAR-100 and Imagenette to test more complex class imbalance patterns (see Section 4.1 of the main text). In particular, we now use a logarithmically decreasing ratio for CIFAR-100 (detailed in Section B.5 of the Appendix) and a linearly decreasing “step-wise” ratio for Imagenette (detailed in Section B.4 of the Appendix). Apart from focal loss-based training, we also now benchmark learn2mix versus SMOTE-based training [2] (a widely used oversampling approach to address class imbalance). Across both CIFAR-100 and Imagenette, we observe that learn2mix accelerates convergence and yields improved performance over classical training, focal loss-based training, and SMOTE-based training. We have provided the complete code in the supplementary materials to reproduce these results (alongside all the results presented in the main text).
>
>
> **References:**
>
> [2] Nitesh V Chawla, Kevin W Bowyer, Lawrence O Hall, and W Philip Kegelmeyer. SMOTE: Synthetic minority over-sampling technique. In Proceedings of the 2002 Joint Conference on IEEE International Conference on Knowledge Discovery and Data Mining and IEEE European Conference on Machine Learning, pp. 878–884. IEEE, 2002.

---

> > ### Author Response · Authors · 2024-11-25
> > **Kind Reminder**
> >
> > Dear Reviewer 5wxS,
> >
> > As we near the end of this discussion period, we wish to extend our sincere gratitude for the time and effort you have invested in reviewing our work.
> >
> > This message serves as a gentle reminder to kindly let us know if you have any further questions we can assist with and if you are considering adjusting your assessment of our work based on the feedback received.
> >
> > Best regards,
> > The Authors

---

### Meta-Review · Area_Chair_PXjD · 2024-12-18

**Metareview:**

This paper proposes Learn2Mix, a novel training strategy that dynamically adjusts the proportion of different classes within training batches based on their error rates. This adaptive approach aims to accelerate model convergence by focusing training efforts on more challenging classes. The paper presents theoretical justification for Learn2Mix and demonstrates its effectiveness on various tasks (classification, regression, reconstruction) and datasets. Nevertheless, according to the reviewers the paper has some weaknesses. They include limited evaluation, with oversimplified class imbalance scenarios in some experiments and
underwhelming performance compared to established benchmarks in certain cases. There is also a lack of comparison with other widely used class imbalance techniques (e.g., oversampling, focal loss). The theoretical results may not fully apply to the complex, non-convex loss landscapes of real-world neural networks. Summing up, the reviewers generally agree that Learn2Mix presents an interesting and potentially valuable approach.  However, they also raise concerns about the evaluation and the applicability of the theoretical results. In general, I believe that this is a border-line paper that requires more work to be accepted. I would suggest that the authors address these concerns and strengthen the experimental evidence.

**Additional Comments On Reviewer Discussion:**

During the discussion, the authors provided feedback about the points of criticism of the reviewers. However, most of their concerns about the paper still remain.

---

### Decision · Program_Chairs · 2025-01-22

Reject